# LIVE: Long-horizon Interactive Video World Modeling

**Junchao Huang** [1 2 3]  **Ziyang Ye** [1]  **Xinting Hu** [4]
**Tianyu He** [3]  **Guiyu Zhang** [1]  **Shaoshuai Shi** [5]  **Jiang Bian** [3]  **Li Jiang** [1 2]

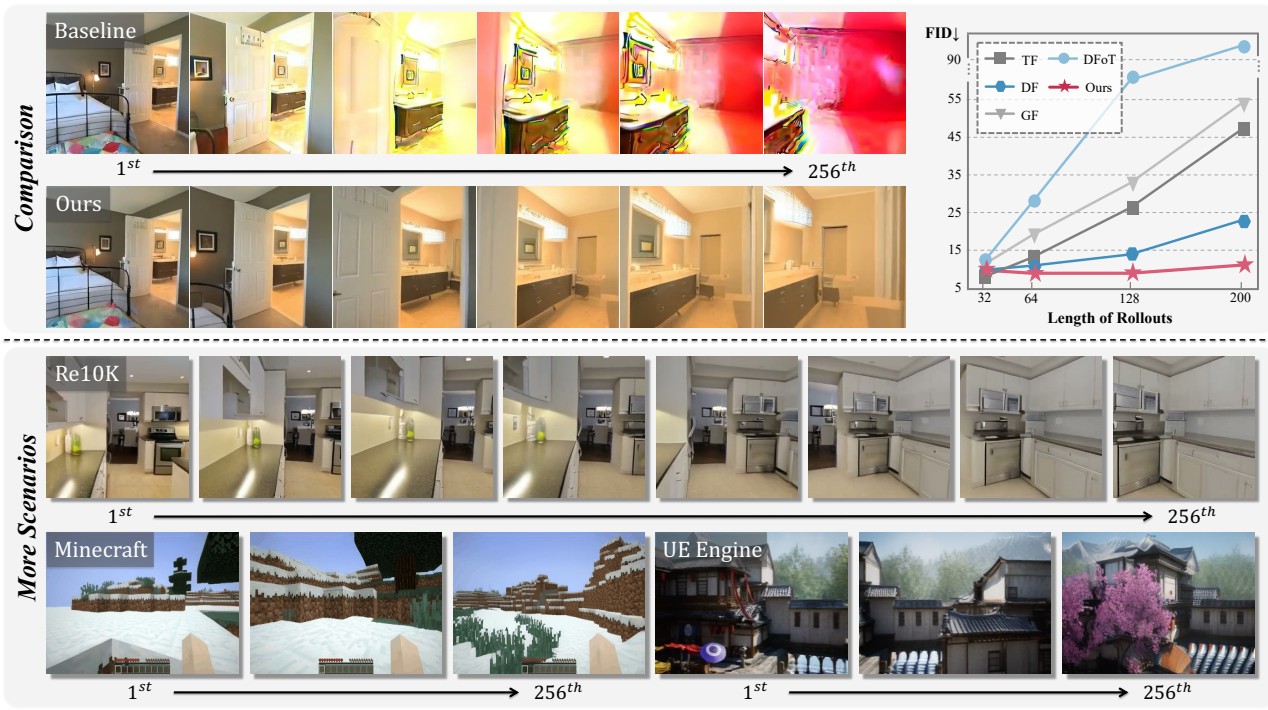

*Figure 1.* LIVE achieves bounded error accumulation for stable long-horizon video world modeling. **Top:** Qualitative comparison with baselines and FID curves showing LIVE maintains stable quality while other methods degrade as rollout length increases. **Bottom:** Applications in real-world (RealEstate10K) and gaming environments (Minecraft, UE Engine). **Project page:** The project page is available at https://junchao-cs.github.io/LIVE-demo/.

## Abstract

Autoregressive video world models predict future visual observations conditioned on actions. While effective over short horizons, these models often struggle with long-horizon generation, as small prediction errors accumulate over time. Prior methods alleviate this by introducing pre-trained teacher models and sequence-level distribution matching, which incur additional computational

cost and fail to prevent error propagation beyond the training horizon. In this work, we propose LIVE, a Long-horizon Interactive Video world modEl that enforces bounded error accumulation via a novel cycle-consistency objective, thereby eliminating the need for teacher-based distillation. Specifically, LIVE first performs a forward rollout from ground-truth frames and then applies a reverse generation process to reconstruct the initial state. The diffusion loss is subsequently computed on the reconstructed terminal state, providing an explicit constraint on long-horizon error propagation. Moreover, we provide a unified view that encompasses different approaches and introduce progressive training curriculum to stabilize training. Experiments demonstrate that LIVE achieves state-of-the-art performance on

[1]The Chinese University of Hong Kong, Shenzhen, Shenzhen, China [2]Shenzhen Loop Area Institute, Shenzhen, China [3]Microsoft Research, Beijing, China [4]The University of Hong Kong, Hong Kong SAR, China [5]Voyager Research, Didi Chuxing, Beijing, China. Correspondence to: Li Jiang <jiangli@cuhk.edu.cn>.

*Proceedings of the 43rd International Conference on Machine Learning*, Seoul, South Korea. PMLR 306, 2026. Copyright 2026 by the author(s).

long-horizon benchmarks, generating stable, high-quality videos far beyond training rollout lengths.

## 1. Introduction

Video world models aim to learn action-conditioned future video predictions for interactive agents, based on past observations and control inputs such as camera poses and keyboard commands. Different from bidirectional video diffusion models that generate the entire video frames at once (OpenAI, 2024; Yang et al., 2024; Polyak et al., 2024; Google, 2025), effective world models require fine-grained interactivity and real-time inference. To achieve this goal, approaches such as Teacher Forcing (TF) (Gao et al., 2024; Hu et al., 2024; Jin et al., 2024) and Diffusion Forcing (DF) (Chen et al., 2024) have introduced causal attention mechanisms into video diffusion models, enabling autoregressive video generation with real-time interactivity.

Despite its empirical success (Zhang et al., 2025a; Chen et al., 2025), autoregressive video world modeling is fundamentally limited by the temporal accumulation of generation errors. This issue arises from exposure bias, where the model is trained on ground-truth frames but must condition on its own predictions at inference time, leading to compounding distributional shift over long horizons. DF (Chen et al., 2024; Song et al., 2025) attempts to mitigate this issue by injecting stochastic noise into the conditioning context during training, thereby exposing the model to imperfect inputs. While this provides some degree of robustness for short sequences, the approach remains ineffective for long-horizon generation, as there remains a substantial distributional gap between noised ground-truth data and genuine model rollouts with accumulated errors.

To further mitigate the train-inference gap, Self-Forcing (SF) (Huang et al., 2025b) has proposed training on rollouts generated by the model itself and distilling knowledge from a pre-trained teacher via holistic sequence-level distribution matching. While effective, this paradigm suffers from several limitations. First, the reliance on pre-trained, interaction-capable teacher models incurs substantial computational overhead, particularly in domain-specific settings. Second, knowledge distillation inherently constrains the student model by the teacher's capacity and can induce mode-seeking behavior that degrades output diversity. Third, SF applies distribution matching at the sequence level without explicitly bounding error accumulation, limiting its ability to control long-horizon error propagation. As a result, the model is only exposed to errors within a fixed training rollout length, and inference beyond this horizon leads to unseen error patterns and potential catastrophic collapse.

To address these limitations, we propose LIVE, a Long-horizon Interactive Video world modEl that enforces bounded error accumulation via a novel cycle-consistency objective, thereby eliminating the reliance on teacher-based distillation. Instead of matching full sequence distributions (Huang et al., 2025b), LIVE performs a forward rollout from ground-truth frames followed by a reverse generation process to reconstruct the initial state, on which the diffusion loss is computed. This formulation explicitly enforces cycle consistency: training the model to map its own imperfect rollouts back to the ground-truth manifold. Crucially, unlike sequence-level objectives that permit unbounded drift, the proposed design maintains distributional alignment between generated rollouts and supervision targets. By training with fixed-length windows while explicitly modeling error accumulation, LIVE learns to operate within a controlled error bound, enabling stable generalization to long-horizon generation at inference time.

In addition, we present a unified view that encompasses TF, DF, and the proposed LIVE. Under this unified view, TF and DF emerge as special cases of LIVE by adjusting the proportion of ground-truth conditioning. Motivated by this observation, we introduce a progressive training curriculum that explicitly controls error tolerance by parameterizing the ratio of ground-truth frames to model-generated rollouts within each training window. This curriculum facilitates stable optimization while preserving high-quality generation through end-to-end diffusion training. In summary, our contributions are threefold:

- We propose LIVE, a long-horizon interactive video world model that enforces bounded error accumulation via a cycle-consistency objective, eliminating the need for teacher-based distillation.
- We present a unified view of TF, DF, and LIVE, and derive a progressive training curriculum that controls error tolerance by adjusting the ratio of ground-truth to rollout frames, enabling stable end-to-end diffusion training.
- We demonstrate state-of-the-art performance on long-horizon interactive video benchmarks, with robust generalization to sequences far beyond the training horizon.

## 2. Related Work

**Video Diffusion Models.** Early video diffusion methods extended image diffusion into temporal domains using UNet-based architectures (Bar-Tal et al., 2024; Blattmann et al., 2023a;b; Guo et al., 2023; Hong et al., 2022). The introduction of Diffusion Transformers (DiT) (Peebles & Xie, 2023; Gupta et al., 2024) enabled better modeling of global spatiotemporal dependencies, leading to large-scale models like Sora (OpenAI, 2024), Seaweed (Seawead et al., 2025), HunyuanVideo (Kong et al., 2024), and Wan (Wan et al., 2025). These bidirectional approaches (Brooks et al., 2024; Bao et al., 2024) employ full-sequence attention where all

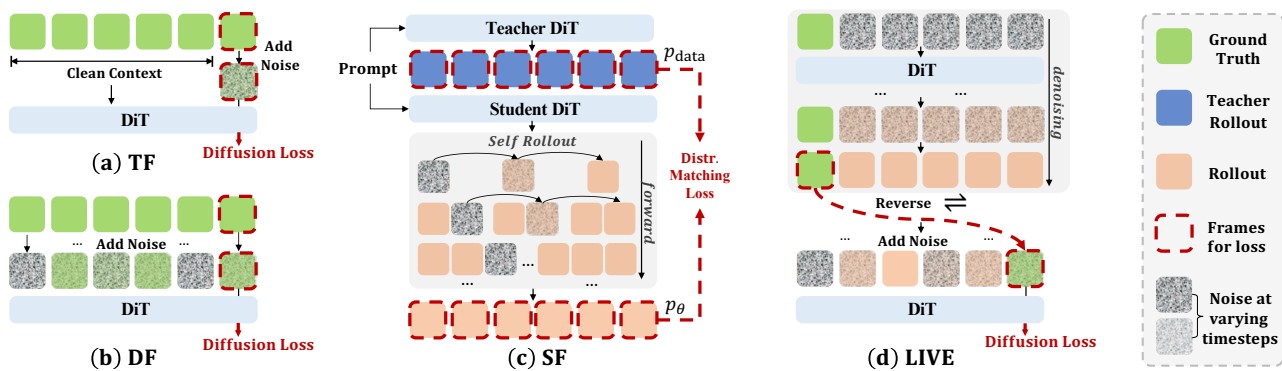

*Figure 2.* Comparison of autoregressive training paradigms. Teacher Forcing (TF) uses ground truth context during training, causing train-inference mismatch. Diffusion Forcing (DF) injects noise but fails to model real rollout errors. Self-Forcing (SF) employs sequence-level distillation with unbounded error accumulation. Our LIVE performs forward rollout then reverse recovery with frame-level diffusion loss, bounding errors through the cycle-consistency objective.

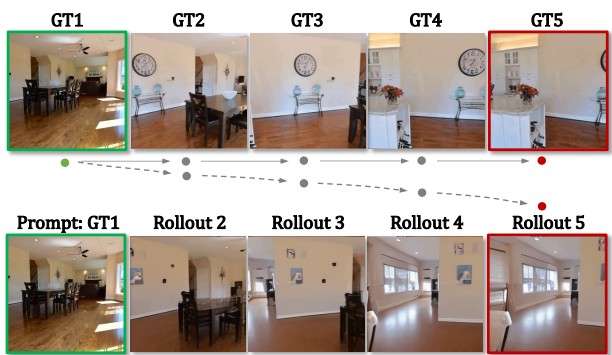

*Figure 3.* Rollout from GT produces semantically diverse content, making direct supervision infeasible. LIVE addresses this by requiring the model to generate back toward the original GT, enabling valid supervision through the cycle-consistency objective.

frames interact simultaneously, achieving impressive temporal consistency and motion quality. However, they are constrained to fixed-length generation and lack frame-level interaction control, while facing computational complexity that scales quadratically with sequence length, making them unsuitable for real-time interactive world modeling.

**Autoregressive Video Generation.** Autoregressive methods synthesize videos sequentially by conditioning on preceding context (Harvey et al., 2022; Li et al., 2024; Xie et al., 2025; Teng et al., 2025; Henschel et al., 2025). Approaches include discrete token-based autoregression (Wu et al., 2024; Kondratyuk et al., 2023; Guo et al., 2025a) and diffusion-based frameworks (Chen et al., 2025; Gu et al., 2025). This paradigm naturally supports interactive world modeling where environments are simulated step-by-step (Feng et al., 2024; Parker-Holder et al., 2024; Valevski et al., 2024; Zhang et al., 2025b; He et al., 2025; Che et al., 2024). Several works adopt causal attention with sliding windows for real-time generation (Decart et al., 2024; Cheng et al., 2025; Huang et al., 2025a), while facing error accu-

mulation challenges during long-horizon inference.

**Mitigating Exposure Bias.** Teacher Forcing (Gao et al., 2024; Jin et al., 2024; Zhang et al., 2025a) conditions on ground truth during training but causes exposure bias at inference when models encounter their own imperfect rollouts. Diffusion Forcing (Chen et al., 2024; Song et al., 2025; Chen et al., 2025; Yin et al., 2025; Gu et al., 2025) injects noise into ground truth context during training to approximate rollout distributions, yet noised ground truth still fundamentally differs from actual rollouts. Self-Forcing (Huang et al., 2025b) and its extensions (Yang et al., 2025; Cui et al., 2025) align the model's rollout distribution with that of a pretrained bidirectional teacher during training, which slows down error accumulation but still suffers from degradation beyond training rollout lengths. Moreover, the reliance on pre-trained teachers complicates extension to interactive video generation models. Concurrent work (Po et al., 2025) constructs corrective trajectories from the model's rollouts to teach it to recover from its mistakes. Another approach (Guo et al., 2025b) simulates rollouts via resampling ground truth, yet this still differs from genuine rollouts.

## 3. Preliminaries

### 3.1. Interactive Video World Modeling

We consider video world modeling as learning the conditional distribution $p(x^{1:T}|c^{1:T})$, where $x^{1:T} = (x^1, \ldots, x^T)$ denotes a sequence of $T$ video frames and $c^{1:T} = (c^1, \ldots, c^T)$ represents conditioning information for each frame (e.g., camera poses, actions).

Video diffusion models learn to denoise Gaussian noise through an iterative process, where a forward diffusion process gradually adds noise to the data:

$$q(x_{t_i}|x_{t_{i-1}}) = \mathcal{N}(x_{t_i}; \sqrt{1-\beta_i}x_{t_{i-1}}, \beta_i I), \quad (1)$$

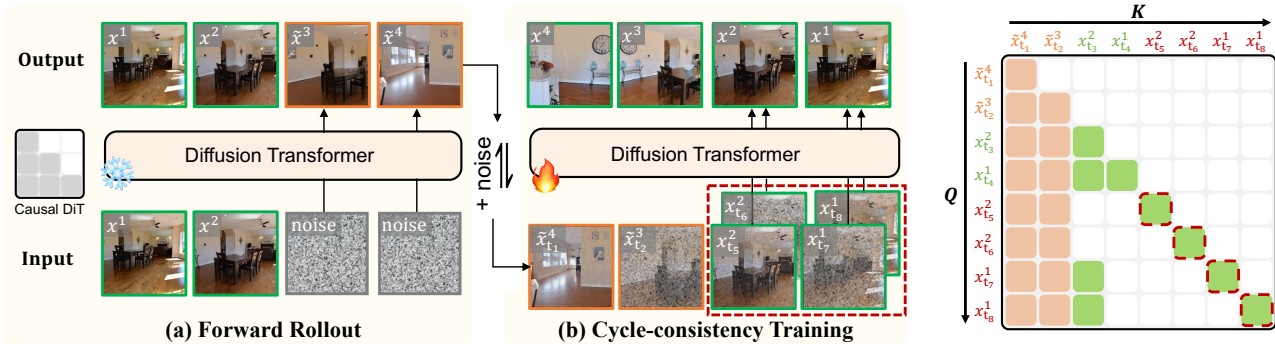

*Figure 4.* LIVE training pipeline. Forward rollout (**Left**, frozen): Given $p$ prompt frames $x^i$, the model generates the remaining $T - p$ frames $\tilde{x}^j$ via causal attention. Cycle-consistency objective (**Right**, trainable): The rollout is reversed and used as context to recover the original prompt frames via frame-level diffusion loss, employing reverse attention (right mask, shown for $p = 2$).

and a reverse denoising process learns to predict the noise:

$$\epsilon_\theta(x_t^k, t, c^k) \approx \epsilon, \qquad (2)$$

where $t \in [t_1, \ldots, t_N]$ denotes the diffusion timestep for frame $k$ and $\epsilon \sim \mathcal{N}(0, I)$ is the Gaussian noise.

In **video world modeling**, the model generates frames sequentially conditioned on previous frames:

$$p(x^{1:T}|c^{1:T}) = \prod_{k=1}^{T} p(x^k|x^{<k}, c^{\leq k}), \qquad (3)$$

where each frame $x^k$ is generated conditioned on the context window $x^{<k} = (x^1, \ldots, x^{k-1})$ and corresponding conditions $c^{\leq k} = (c^1, \ldots, c^k)$. For interactive world models requiring real-time inference, we employ a **sliding window** approach where only the most recent $K$ frames are used as context:

$$p(x^k|x^{k-K:k-1}, c^{k-K:k}). \qquad (4)$$

### 3.2. Training Paradigms for Autoregressive Generation

Existing autoregressive video diffusion models typically employ one of three training strategies (**Figure 2** (a)-(c)):

**Teacher Forcing (TF).** During training, the model predicts noise conditioned on ground truth frames within a sliding window:

$$\mathcal{L}_{\text{TF}} = \mathbb{E}_{\substack{x^{1:K}, \\ \epsilon, t_i}} \left[ \sum_{k=1}^{K} \left\| \epsilon^k - \epsilon_\theta(x_{t_i}^k, x^{<k}, t_i, c^{\leq k}) \right\|^2 \right]. \qquad (5)$$

where $x_{t_i}^k = \alpha_{t_i} x^k + \sigma_{t_i} \epsilon^k$ is the noised frame $k$ at timestep $t_i$, with $t_i$ independently sampled for each frame from the noise schedule $[t_1, \ldots, t_N]$, and $K$ denotes the context window length. This creates a train-inference discrepancy: at inference, the model must condition on its own imperfect rollouts rather than ground truth.

**Diffusion Forcing (DF).** To bridge this gap, DF injects noise into the conditioning context during training:

$$\mathcal{L}_{\text{DF}} = \mathbb{E}_{\substack{x^{1:K}, \\ \epsilon, t_i}} \left[ \sum_{k=1}^{K} \left\| \epsilon^k - \epsilon_\theta(x_{t_i}^k, \hat{x}^{<k}, t_i, c^{\leq k}) \right\|^2 \right], \qquad (6)$$

where $\hat{x}^j = \alpha_{t_i} x^j + \sigma_{t_i} \epsilon^j$ represents noisy context frame $j$ with independently sampled timestep $t_i$, and $\epsilon^j \sim \mathcal{N}(0, I)$. However, the distribution of noised ground truth differs from genuine model rollouts with accumulated errors.

**Self-Forcing (SF).** SF addresses this through knowledge distillation, where a student model learns from its own rollouts under the supervision of a teacher:

$$\mathcal{L}_{\text{SF}} = D_{KL}\left(p_{\text{teacher}}(\tilde{x}^{1:T}) \| p_{\text{student}}(\tilde{x}^{1:T})\right). \qquad (7)$$

However, sequence-level distribution matching fails to constrain error accumulation within bounded ranges, leading to quality degradation that prevents generalization beyond training rollout lengths.

## 4. Method

We introduce LIVE, a framework that enforces bounded error accumulation via a cycle-consistency constraint. Specifically, LIVE performs a forward rollout from ground-truth (GT) frames followed by a reverse generation process to reconstruct the initial state, on which the diffusion loss is computed. This formulation explicitly enforces cycle consistency by training the model to map its own imperfect rollouts back to the GT manifold.

### 4.1. Bounded Error Accumulation

Consider an autoregressive video diffusion model that generates frames sequentially: $p(x^{1:T}|c^{1:T}) = \prod_{k=1}^{T} p_\theta(x^k|x^{<k}, c^{\leq k})$, where $x^{<k} = (x^1, \ldots, x^{k-1})$ denotes the context frames and $c^{1:T}$ represents conditioning information (e.g., camera poses, actions).

**Algorithm 1** LIVE Training Pipeline

**Require:** Window length $T$, minimum $p_{\min}$
1: **for** each epoch **do**
2:     Pre-training: $p \leftarrow T$
3:     Post-training: decrease $p$ gradually
4:     **for** each batch $(x^{1:T}, c^{1:T}) \in \mathcal{D}$ **do**
5:         $\tilde{x}^{p+1:T} \sim p_\theta(x^{p+1:T}|x^{1:p}, c^{1:T})$ {Eq. 9}
6:         $\tilde{x}^{p+1:T,\text{rev}} \leftarrow (\tilde{x}^T, \dots, \tilde{x}^{p+1})$ {Eq. 10}
7:         $c^{\text{rev}} \leftarrow (c^T, \dots, c^1)$ {Eq. 10}
8:         Inject noise: $\tilde{x}^{k,\epsilon} \leftarrow \alpha_t \tilde{x}^k + \sigma_t \eta^k$ {Eq. 11}
9:         Compute $\mathcal{L}_{\text{LIVE}}$ {Eq. 13}
10:       $\theta \leftarrow \theta - \alpha \nabla_\theta \mathcal{L}_{\text{LIVE}}$
11:     **end for**
12: **end for**

**Problem Setup.** During autoregressive generation with rollouts, we observe a general tendency toward quality degradation in expectation:

$$\mathbb{E}[\mathcal{D}(x^k, \tilde{x}^k)] \lesssim \mathbb{E}[\mathcal{D}(x^{k+1}, \tilde{x}^{k+1})], \quad \forall k \in [1, T-1], \tag{8}$$

where $\mathcal{D}(x^k, \tilde{x}^k)$ measures perceptual quality (e.g., FVD, FID). While this error accumulation pattern is empirically well-established, directly supervising rollouts to reduce $\mathcal{D}(x^k, \tilde{x}^k)$ faces a fundamental obstacle: rollouts naturally produce semantically diverse content that diverges from GT trajectories (**Figure 3**). Since $\tilde{x}^k$ and $x^k$ represent different but equally valid future states, computing diffusion loss between them is infeasible. This limitation hinders extending SF to efficient parallel diffusion supervision and increases its dependence on pretrained teacher models.

**Cycle-consistency Objective.** To address the above challenges, LIVE introduces a cycle-consistency objective that enables valid frame-level supervision without requiring distributional alignment between rollouts and GT. The key insight is: instead of supervising rollouts $\tilde{x}^{p+1:T}$ (where $p$ denotes the number of prompt frames used to initiate the rollout) directly against GT, we require them to be *recoverable* - the model must be able to reverse-generate the original GT prompt frames from the rollouts by reversing camera poses/actions. This creates a valid training signal while accommodating distributional diversity. For a video sequence $x^{1:T}$:

*Step 1 (Forward Rollout):* Given a training window of $T$ frames with known camera/action conditions, we use the first $p$ frames as prompt frames and generate the remaining $T - p$ frames with gradients disabled. Unlike inference which requires frame-by-frame interaction, during training we can efficiently generate all $T - p$ frames simultaneously (initialized from pure noise) since we have access to all future camera/action conditions. This uses the same causal attention mask as inference (**Figure 4**), ensuring training-

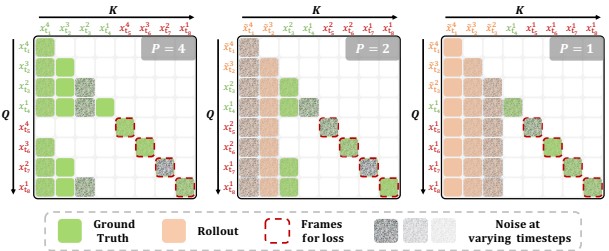

*Figure 5.* Progressive training curriculum by increasing rollout ratio. From left to right, as $p$ decreases, more generated frames enter the context, increasing the model's error tolerance while maintaining recoverability through the cycle-consistency objective.

inference consistency while dramatically improving efficiency:

$$\tilde{x}^{p+1:T} \sim p_\theta(x^{p+1:T}|x^{1:p}, c^{1:T}). \tag{9}$$

*Step 2 (Reverse Generation):* Reverse the rollout temporally and reverse camera/action conditions, then attempt to recover the original $p$ prompt frames. Since forward rollouts satisfy $\mathcal{D}(\tilde{x}^k, x^k) \leq \mathcal{D}(\tilde{x}^{k+1}, x^{k+1})$ (quality degrades monotonically), after reversal the context quality *improves* monotonically. Without intervention, the model could recover $x^1$ by attending primarily to the highest-quality context frame $\tilde{x}^{2,\text{rev}}$, trivially satisfying recoverability without constraining forward errors. To prevent this shortcut, we first reverse the rollout:

$$\tilde{x}^{p+1:T,\text{rev}} \leftarrow (\tilde{x}^T, \dots, \tilde{x}^{p+1}), \; c^{\text{rev}} \leftarrow (c^T, \dots, c^1), \tag{10}$$

then inject random noise per frame. For each $k \in [p+1, T]$, sample $t \sim \mathcal{U}([t_1, \dots, t_N])$ and $\eta^k \sim \mathcal{N}(0, I)$, then:

$$\tilde{x}^{k,\epsilon} \leftarrow \alpha_t \tilde{x}^k + \sigma_t \eta^k, \tag{11}$$

and finally recover the original prompts:

$$\hat{x}^{1:p} \sim p_\theta(x^{1:p}| \; \tilde{x}^{p+1:T, \text{rev}, \epsilon}, \; c^{\text{rev}}). \tag{12}$$

*Step 3 (Frame-Level Supervision):* To enable efficient parallel training like TF/DF, we extend the $p$-frame supervision to the full window length $T$ by repeating the $p$ GT frames and applying different noise timesteps to each position. This allows computing noise prediction loss on all $T$ frames in parallel:

$$\mathcal{L}_{\text{LIVE}} = \mathbb{E}_{\substack{x^{1:T} \sim p_{\text{data}} \\ t \sim \mathcal{U}([t_1, \dots, t_N]) \\ \epsilon^k \sim \mathcal{N}(0, I)}} \left[ \frac{1}{T} \sum_{k=1}^{T} \left\| \epsilon^k - \epsilon_\theta^k \right\|^2 \right], \tag{13}$$

where each $\epsilon_\theta^k$ is predicted as:

$$\epsilon_\theta^k = \epsilon_\theta(x_t^{\text{gt}(k)}, \tilde{x}^{<k,\text{rev},\epsilon}, t, c^{\leq k}), \tag{14}$$

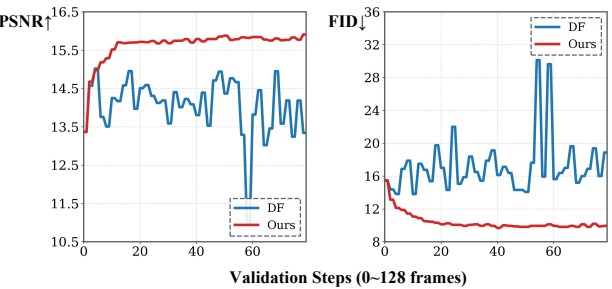 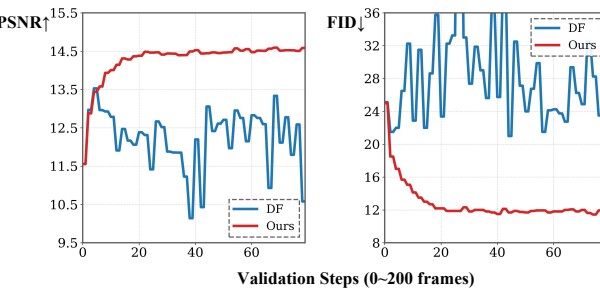

*Figure 6.* Post-training performance from a converged DF checkpoint. Continued DF training stagnates with oscillating metrics, while LIVE achieves substantial improvements that amplify at longer horizons. LIVE converges to comparable FID across 128-frame and 200-frame generation, demonstrating uniform quality regardless of rollout length.

with $t$ independently sampled for each frame $k$ from the noise schedule, $x^{\text{gt}(k)}$ denoting the GT frame (repeated from the $p$ prompts), and $\tilde{x}^{<k,\text{rev},\epsilon}$ representing the reversed rollout context with injected noise. Specifically, $\tilde{x}^{<k,\text{rev},\epsilon}$ consists of the subset of frames from $\tilde{x}^{p+1:T,\text{rev},\epsilon}$ in **Eq. 12** preceding position $k$.

**Implicit Error Bounding.** The cycle-consistency objective creates an implicit incentive to bound forward distortion through the recovery objective. Define $\mathcal{D}_{\text{ctx}} = \mathcal{D}(x^k, \tilde{x}^k)$ as the distortion between rollout and GT at frame $k$, and $\mathcal{D}_{\text{rec}} = \frac{1}{p} \sum_{k=1}^{p} \mathcal{D}(x^k, \hat{x}^k)$ as the average recovery distortion over the $p$ prompt frames. The training objective minimizes $\mathcal{D}_{\text{rec}}$, which encourages:
(1) Maintaining forward distortion $\mathcal{D}(x^k, \tilde{x}^k)$ within bounded range to enable recovery from rollout context;
(2) Optimizing $\epsilon_\theta$ to recover GT frames from imperfect rollout context, directly reducing $\mathcal{D}_{\text{rec}}$ via gradient descent.

Consequently, gradient optimization learns to maintain $\mathcal{D}(x^k, \tilde{x}^k)$ within a bounded range, preventing the monotonic degradation that plagues AR diffusion inference.

### 4.2. Progressive Training Curriculum

**Unified Training Objective.** LIVE unifies existing training paradigms by controlling the GT ratio $p \in [1, T]$, encompassing: (1) Teacher Forcing ($p = T$, perfect GT context $x^{<k}$); (2) Diffusion Forcing ($p = T$, noisy GT context $\hat{x}^{<k} = x^{<k} + \epsilon$); (3) LIVE ($p < T$, imperfect rollout context $\tilde{x}^{<k}$ with accumulated errors). By controlling $p$, our framework supports both pre-training and post-training: during pre-training, the model uses $p = T$ since it has not yet learned to generate rollouts; during post-training, as shown in **Figure 5**, we progressively decrease $p$ to adapt the model to increasing error levels.

**Error Tolerance.** When $p$ is large, the context consists primarily of GT frames with small distortions, making recovery relatively easy; as $p$ decreases, more rollout frames enter the context, accumulating larger errors and making recovery increasingly difficult. Through gradually exposing the

model to harder recovery tasks, this strengthens its ability to recover from imperfect contexts (Error Tolerance). This enhanced capability, in turn, produces better rollouts with reduced errors, enabling robust long-horizon generation.

## 5. Experiments

**Implementation Details.** All experiments are conducted on a cluster of 32 NVIDIA H100 GPUs with a batch size of 64. Our model architecture follows the NFD (Cheng et al., 2025) 774M configuration. For RealEstate10K, we train from scratch following DFoT (Song et al., 2025) settings at $256 \times 256$ resolution with a frame skip of 2. For UE Engine Videos datasets (Yu et al., 2025), we initialize from RealEstate10K pre-trained weights and apply the same frame skip of 2. Our model use a fixed context window of 32 frames during both training and evaluation. Additional implementation details are provided in Appendix A.1.

**Datasets and Baselines.** We evaluate on three diverse benchmarks: (1) RealEstate10K: A large-scale dataset of real estate videos featuring diverse camera motions. We report results on the complete test set. (2) UE Engine Videos: Following Context-as-Memory (Yu et al., 2025), we use their dataset containing 100 videos of 7,601 frames across 12 scenes with camera pose annotations, collected from realistic game engine environments (UE engine). We randomly select one video per scene (12 videos total) as the test set. (3) Minecraft: We train on the WorldMem (Xiao et al., 2025) dataset and collect 300 video-action pairs from MineDojo (Fan et al., 2022) for evaluation, testing long-horizon generation in interactive environments. We compare LIVE against multiple baselines including CameraCtrl (He et al., 2024), DFoT (Song et al., 2025), Geometry Forcing (Wu et al., 2025), and NFD-TF/DF (Teacher Forcing/Diffusion Forcing) (Cheng et al., 2025), assessing generation quality using PSNR, SSIM, LPIPS, and FID metrics. We focus our comparison on methods without interactive teacher model distillation. Training large-scale bidirectional teacher models (Huang et al., 2025b) remains important future work beyond our current computational budget.

*Table 1.* RealEstate10K full test set results across different rollout lengths. LIVE achieves state-of-the-art performance, with particularly large gains at longer sequences demonstrating superior long-horizon generation capability.

| Method | Real-time | 0∼64 frames | | | 0∼128 frames | | | 0∼200 frames | | | ≥256 frames | | |
|---|---|---|---|---|---|---|---|---|---|---|---|---|---|
| | | PSNR ↑ | LPIPS ↓ | SSIM ↑ | PSNR ↑ | LPIPS ↓ | SSIM ↑ | PSNR ↑ | LPIPS ↓ | SSIM ↑ | PSNR ↑ | LPIPS ↓ | SSIM ↑ |
| CameraCtrl (He et al., 2024) | ✗ | 14.09 | 0.3829 | 0.4366 | 11.69 | 0.5224 | 0.3651 | 10.25 | 0.6115 | 0.3181 | 9.48 | 0.6585 | 0.2886 |
| DFoT (Song et al., 2025) | ✗ | 15.65 | 0.3053 | 0.4989 | 12.55 | 0.4601 | 0.3936 | 10.86 | 0.5613 | 0.3287 | 10.02 | 0.6128 | 0.2921 |
| GF (Wu et al., 2025) | ✗ | 16.37 | 0.2450 | 0.5567 | 12.69 | 0.4190 | 0.4534 | 10.59 | 0.5400 | 0.3969 | 9.91 | 0.5936 | 0.3796 |
| NFD-TF (Cheng et al., 2025) | ✓ | 16.87 | 0.2571 | 0.5503 | 13.59 | 0.4302 | 0.4448 | 11.63 | 0.5526 | 0.3724 | 10.58 | 0.6222 | 0.3281 |
| NFD-DF (Cheng et al., 2025) | ✓ | 16.59 | 0.2558 | 0.5723 | 13.82 | 0.3922 | 0.5015 | 12.21 | 0.4956 | 0.4598 | 11.51 | 0.5506 | 0.4397 |
| **LIVE** | ✓ | **18.11** | **0.2215** | **0.5810** | **15.91** | **0.3298** | **0.5096** | **14.57** | **0.4163** | **0.4630** | **13.89** | **0.4682** | **0.4400** |

*Table 2.* Results on interactive game environments. LIVE achieves consistent improvements over baselines on both realistic game engine videos (UE Engine) and interactive gameplay (Minecraft), demonstrating strong performance for interactive world modeling.

| Method | 0∼64 frames | | | 0∼128 frames | | | 0∼256 frames | | | ≥400 frames | | |
|---|---|---|---|---|---|---|---|---|---|---|---|---|
| | PSNR ↑ | LPIPS ↓ | SSIM ↑ | PSNR ↑ | LPIPS ↓ | SSIM ↑ | PSNR ↑ | LPIPS ↓ | SSIM ↑ | PSNR ↑ | LPIPS ↓ | SSIM ↑ |
| *UE Engine (Realistic Game Engine)* | | | | | | | | | | | | |
| NFD-TF (Cheng et al., 2025) | 17.16 | 0.3387 | 0.4953 | 14.95 | 0.4597 | 0.4245 | 12.97 | 0.5702 | 0.3625 | 11.80 | 0.6318 | 0.3286 |
| NFD-DF (Cheng et al., 2025) | 17.15 | 0.3357 | 0.5062 | 14.71 | 0.4586 | 0.4441 | 12.27 | 0.5799 | 0.3956 | 11.02 | 0.6456 | 0.3760 |
| **LIVE** | **17.83** | **0.3145** | **0.5204** | **15.85** | **0.4210** | **0.4600** | **14.04** | **0.5214** | **0.4085** | **12.96** | **0.5794** | **0.3834** |

| Method | 0∼32 frames | | | 0∼64 frames | | | 0∼128 frames | | | 0∼200 frames | | |
|---|---|---|---|---|---|---|---|---|---|---|---|---|
| | PSNR ↑ | LPIPS ↓ | SSIM ↑ | PSNR ↑ | LPIPS ↓ | SSIM ↑ | PSNR ↑ | LPIPS ↓ | SSIM ↑ | PSNR ↑ | LPIPS ↓ | SSIM ↑ |
| *Minecraft (Interactive Gameplay)* | | | | | | | | | | | | |
| NFD-TF (Cheng et al., 2025) | 16.09 | 0.3474 | 0.6224 | 14.62 | 0.4067 | 0.5930 | 13.06 | 0.4781 | 0.5560 | 12.10 | 0.5255 | 0.5311 |
| NFD-DF (Cheng et al., 2025) | 17.39 | 0.2888 | 0.6401 | 15.54 | 0.3586 | 0.6036 | 13.52 | 0.4469 | 0.5594 | 12.34 | 0.5091 | 0.5332 |
| **LIVE** | **17.87** | **0.2698** | **0.6558** | **16.31** | **0.3271** | **0.6291** | **14.90** | **0.3877** | **0.6037** | **14.02** | **0.4299** | **0.5885** |

*Table 3.* Ablation studies on RealEstate10K test set evaluating the impact of key components in LIVE.

| Variant | 0∼64 frames | | | 0∼200 frames | | |
|---|---|---|---|---|---|---|
| | PSNR ↑ | LPIPS ↓ | SSIM ↑ | PSNR ↑ | LPIPS ↓ | SSIM ↑ |
| *Effect of Cycle-consistency Objective* | | | | | | |
| w/o Cycle | 13.99 | 0.4041 | 0.4597 | 11.18 | 0.6024 | 0.3564 |
| *Effect of Context Noise Strategy* | | | | | | |
| No Noise | 17.76 | 0.2310 | 0.5752 | 13.83 | 0.4573 | 0.4487 |
| Fixed Noise | 17.48 | 0.2392 | 0.5551 | 14.09 | 0.4444 | 0.4508 |
| *Effect of Progressive Training Curriculum* | | | | | | |
| Fixed $p = 1$ | 16.78 | 0.2747 | 0.5265 | 13.58 | 0.4800 | 0.4279 |
| **LIVE** | **18.11** | **0.2215** | **0.5810** | **14.57** | **0.4163** | **0.4630** |

## 5.1. Main Results

**Error Accumulation Analysis. Figure** 1 demonstrates LIVE's core advantage. Training models on RealEstate10K with TF (Teacher Forcing), DF (Diffusion Forcing), DFoT (Song et al., 2025), GF (Geometry Forcing) (Wu et al., 2025), and LIVE (TF, DF, and LIVE use the same model architecture (Cheng et al., 2025)), we evaluate FID at 32, 64, 128, and 200 frames. LIVE maintains stable FID around 10 across all lengths, while all baselines degrade dramatically beyond 64 frames, validating that our cycle-consistency objective successfully bounds error accumulation.

**Figure** 6 shows post-training from a converged DF checkpoint. Continued DF training stagnates with oscillating metrics, while LIVE achieves substantial gains that amplify at longer sequences. Critically, LIVE converges to comparable FID for both 64-frame and 200-frame generation, maintaining uniform quality across rollout horizons.

**Quantitative Results.** Tables 1 and 2 show that LIVE achieves substantial improvements over all baselines across three benchmarks, with particularly large gains at longer rollout lengths. Our method shares identical architecture and inference procedures with NFD (Cheng et al., 2025), isolating training strategy as the sole differentiator. While DF improves upon TF by injecting noise during training, it remains insufficient for long-horizon generation since noised ground truth fails to match the distribution of genuine rollouts with accumulated errors. LIVE addresses this limitation by training directly on imperfect rollouts with the cycle-consistency objective, achieving bounded error accumulation through end-to-end diffusion optimization.

**Qualitative Results. Figures** 8 and 7 present qualitative comparisons on UE Engine and RealEstate10K datasets. On UE Engine, we compare models with identical architecture trained using TF, DF, and LIVE, demonstrating LIVE's superior generation quality. On RealEstate10K, our method maintains consistent visual quality over extended rollouts

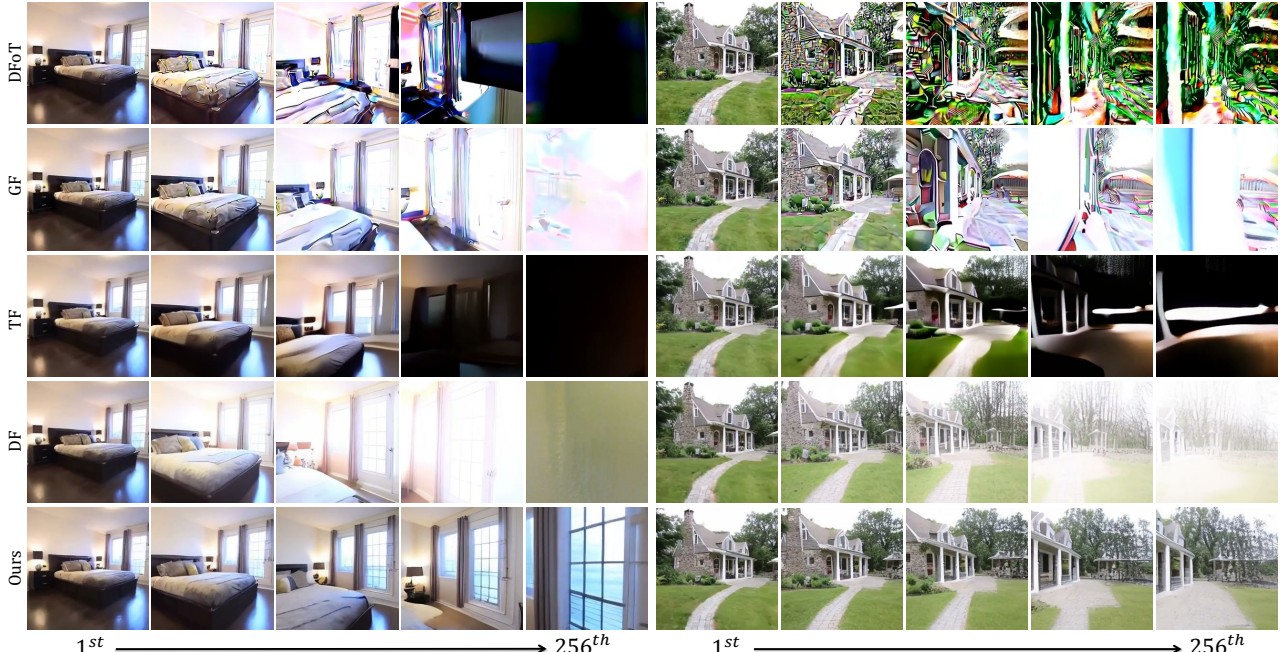

*Figure 7.* Qualitative comparison on RealEstate10K dataset. We showcase indoor and outdoor scenes comparing various methods. LIVE demonstrates stable visual quality during rollouts. Full videos and additional examples are provided in Appendix A.2.

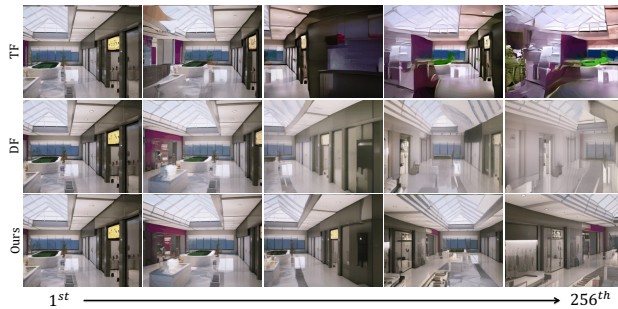

*Figure 8.* Qualitative comparison on UE Engine dataset. We compare models with identical architecture trained using Teacher Forcing (TF), Diffusion Forcing (DF), and LIVE.

across both indoor and outdoor scenes, while competing methods exhibit noticeable degradation. Full videos and additional examples are provided in Appendix A.2.

### 5.2. Ablation Studies

We conduct comprehensive ablation studies on the RealEstate10K test set to validate the key design choices in LIVE. Results are summarized in Table 3.

**Effect of Cycle-consistency Objective.** Removing the reverse generation step leads to substantial performance degradation. This validates our core hypothesis illustrated in **Figure** 3: direct supervision on forward rollouts is infeasible due to semantic divergence between rollouts and ground truth. The cycle-consistency objective addresses this by

requiring the model to generate back toward the original GT, creating a valid training signal that accommodates distributional diversity while constraining error accumulation within recoverable limits.

**Effect of Context Noise Strategy.** We compare three noise injection strategies for the rollout context: (1) no noise, (2) fixed-scale noise, and (3) random timestep sampling (LIVE). Without noise, the model shows acceptable short-horizon performance but degrades significantly at longer sequences. Fixed-scale noise provides marginal improvement, while our random timestep sampling achieves the best results. This validates the analysis in Step 2 in Sec 4.1: after reversing, context quality improves monotonically, allowing trivial recovery by attending to higher-quality neighboring frame. Random per-frame noise breaks this pattern, forcing model to learn robust recovery from diverse error distributions.

**Effect of Progressive Training Curriculum.** Directly setting $p = 1$ throughout post-training underperforms our progressive curriculum that gradually decreases $p$ from $T$ to $p_{\min}$. Abruptly exposing the model to the maximum rollout length creates an overly difficult task before sufficient error tolerance develops. Progressive rollout extension allows gradual capability building, starting from easy recovery with mostly GT context, then progressively increasing the rollout proportion to expose harder error patterns. This enhanced recovery capability produces better rollouts, ultimately enabling the model's error tolerance to converge smoothly toward its recovery capacity.

# 6. Conclusion

In this work, we introduce LIVE, a long-horizon interactive video world model that addresses the fundamental challenge of error accumulation in autoregressive generation. By enforcing a cycle-consistency objective through diffusion loss, LIVE explicitly bounds long-horizon error propagation without relying on teacher-based distillation. We further present a unified perspective that connects TF, DF, and LIVE, and derived a progressive training curriculum that stabilizes optimization while preserving generation quality. Extensive experiments demonstrate that LIVE achieves strong performance and robust generalization on long-horizon interactive video world modeling benchmarks, significantly extending the effective rollout horizon beyond the training window. In future work, we will further scale up LIVE on large-scale and diverse datasets.

## Impact Statement

This paper presents work whose goal is to advance the field of Machine Learning. There are many potential societal consequences of our work, none of which we feel must be specifically highlighted here.

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

# A. Appendix

## A.1. Implementation Training Details

### A.1.1. REALESTATE10K

Our model has 774M parameters, sharing identical architecture (DiTs) and inference procedures with NFD (Cheng et al., 2025). Specifically, we use 18-step ODE sampling during inference for all methods. The model operates at $256 \times 256$ resolution with a frame skip of 2 during both training and inference, employing the same VAE as NFD for 16× spatial downsampling to the latent space.

All experiments are conducted on a cluster of 32 NVIDIA H100 GPUs with a batch size of 64. We use the Adam optimizer with a learning rate of $4 \times 10^{-5}$. Both NFD-TF and NFD-DF are trained from scratch for over 200k iterations until convergence. Our method (LIVE) is initialized from the converged NFD-DF checkpoint (200k steps) and trained for an additional 20k iterations until convergence.

We use a fixed context window of 32 frames during both training and evaluation. The training set follows DFoT, containing approximately 50-60k videos. All metrics are reported on the complete RealEstate10K test set with over 7k videos.

### A.1.2. UE ENGINE VIDEOS

For UE Engine Videos dataset (Yu et al., 2025), we use the same model configuration as RealEstate10K (774M parameters). We initialize from RealEstate10K ($256 \times 256$) pre-trained weights and fine-tune at $352 \times 640$ resolution with a frame skip of 2. The dataset contains 100 videos totaling 7,601 frames across 12 scenes with camera pose annotations. We randomly select 12 videos (one per scene) for testing and use the remaining 88 videos for training. For evaluation, we uniformly sample 50 starting frames from each test video, resulting in 600 test sequences in total.

All experiments are conducted on 32 NVIDIA H100 GPUs with a batch size of 64. We use the Adam optimizer with a learning rate of $4 \times 10^{-5}$. NFD-TF and NFD-DF are initialized from their respective RealEstate10K checkpoints (TF and DF) and fine-tuned for 10k iterations. Our method (LIVE) is initialized from the RealEstate10K DF checkpoint, first fine-tuned with DF for 10k iterations, then further trained with LIVE for 6.5k iterations. We use the same VAE as RealEstate10K for 16× spatial downsampling to the latent space.

### A.1.3. MINECRAFT

For Minecraft, we use the same model configuration as RealEstate10K (774M parameters) and operate at $224 \times 384$ resolution with a frame skip of 1. The WorldMem (Xiao et al., 2025) training dataset contains approximately 10k interactive gameplay videos of 1500 frames each, collected through MineDojo (Fan et al., 2022), where each frame is accompanied by a 25-dimensional action vector. Since WorldMem does not provide an official test set, we collect 300 action trajectories from MineDojo for evaluation, with each trajectory representing randomly generated gameplay data.

All experiments are conducted on 32 NVIDIA H100 GPUs with a batch size of 64. We use the Adam optimizer with a learning rate of $4 \times 10^{-5}$. NFD-DF is initialized from the original NFD checkpoint (200k steps) and fine-tuned on WorldMem for 30k iterations. NFD-TF is trained from scratch on WorldMem for 100k iterations. Our method (LIVE) is initialized from the converged NFD-DF checkpoint (after 30k iterations on WorldMem) and further trained with LIVE for 3k iterations. We use the same VAE as NFD with the decoder fine-tuned on Minecraft scenarios for 16× spatial downsampling to the latent space.

## A.2. Additional Qualitative Results

We provide additional qualitative examples across different datasets. Through these examples, we observe that different models exhibit distinct failure patterns during long rollouts. For instance, TF models tend to develop color distortion and semantic inconsistency, while DF models show exposure problems with overexposed or underexposed regions. Our method addresses these issues by training the model to recover from its own generated errors, thereby achieving stable generation quality even over extended sequences. The reversibility constraint ensures that the model learns to maintain quality within a bounded range throughout the generation process.

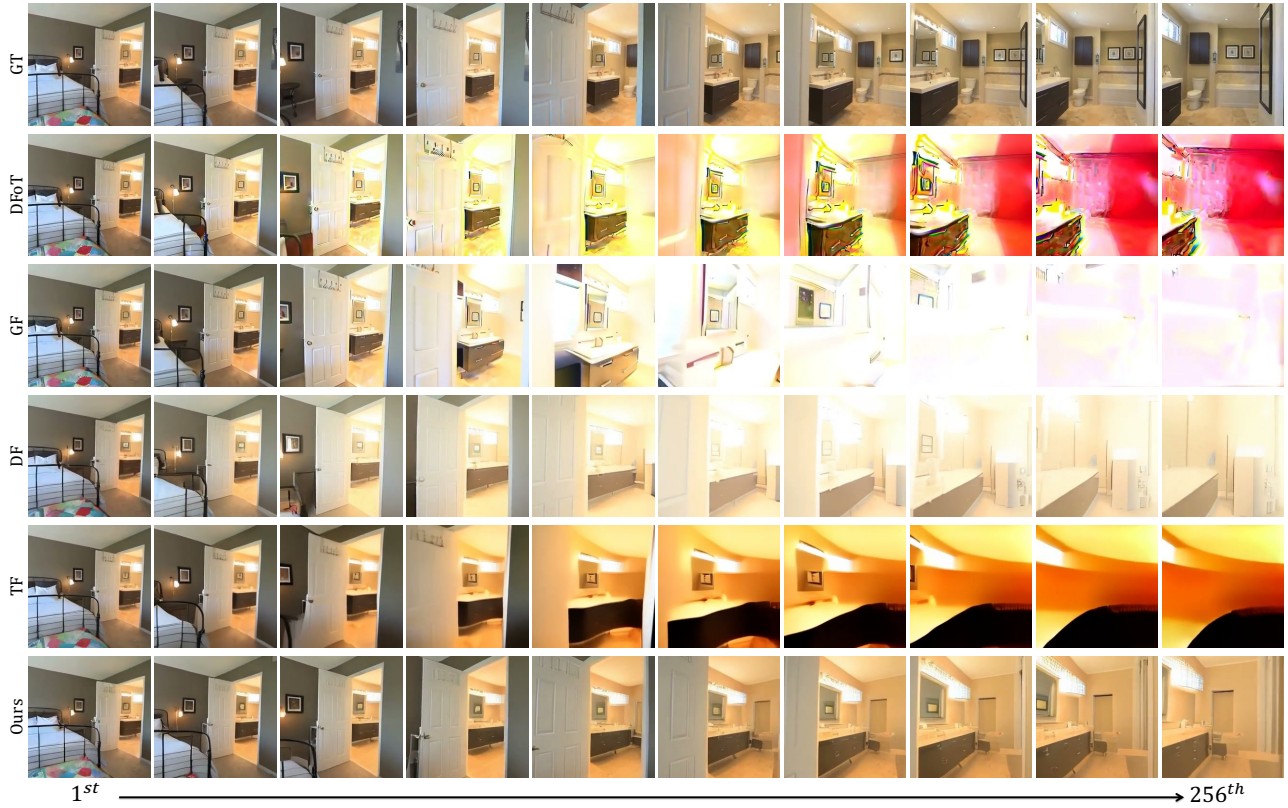

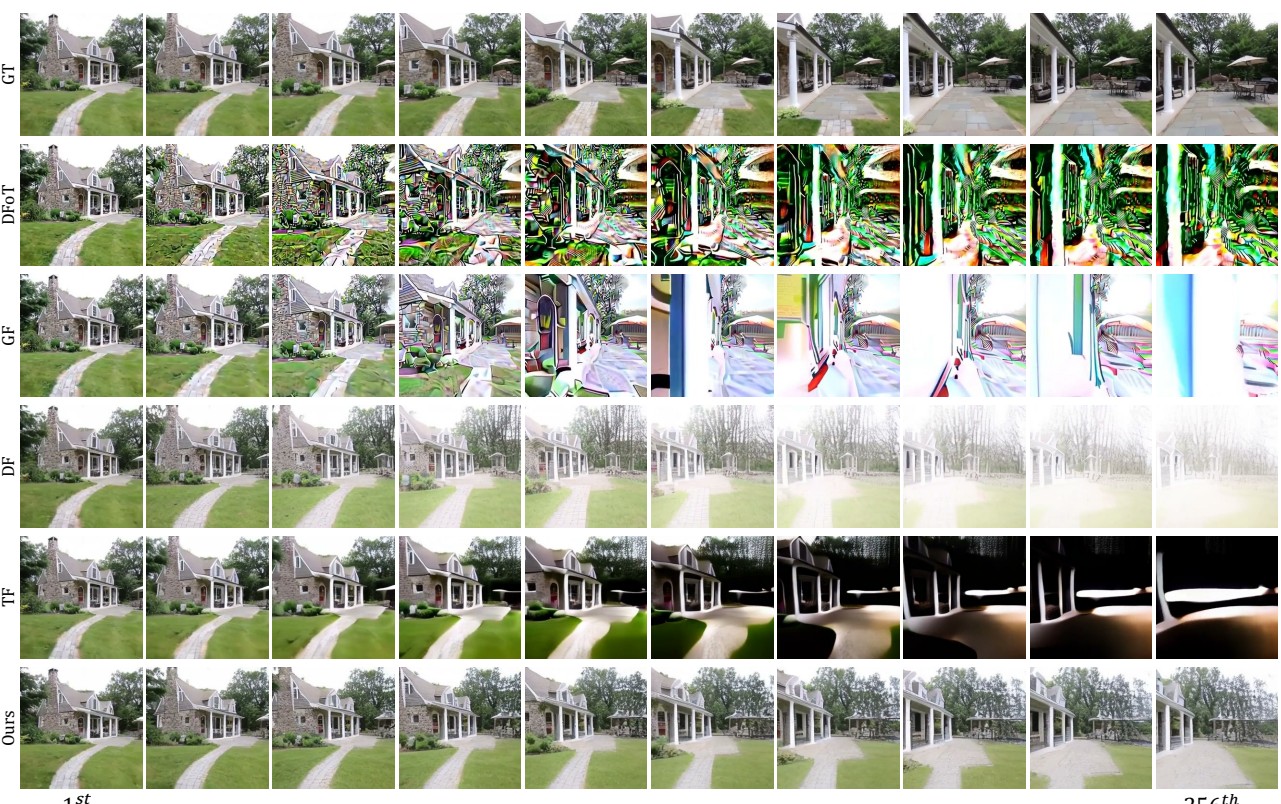

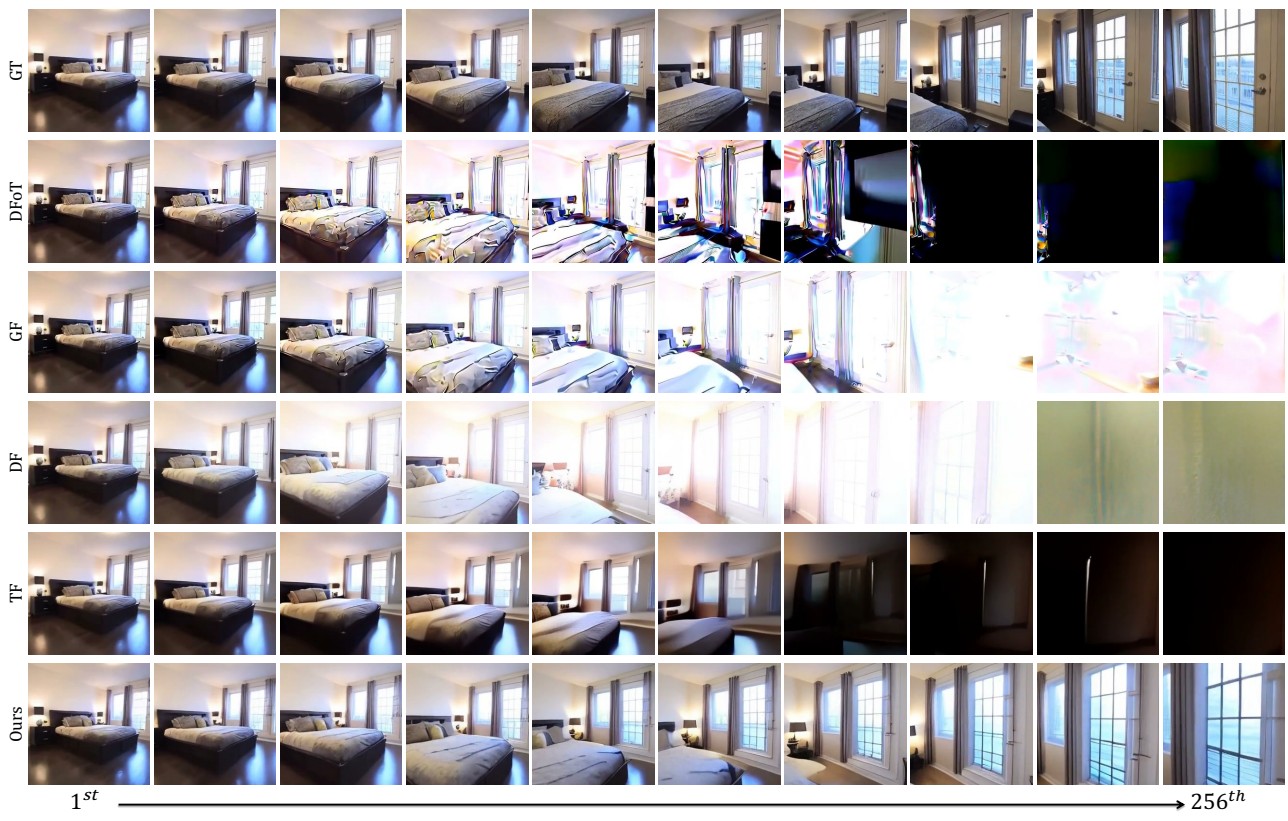

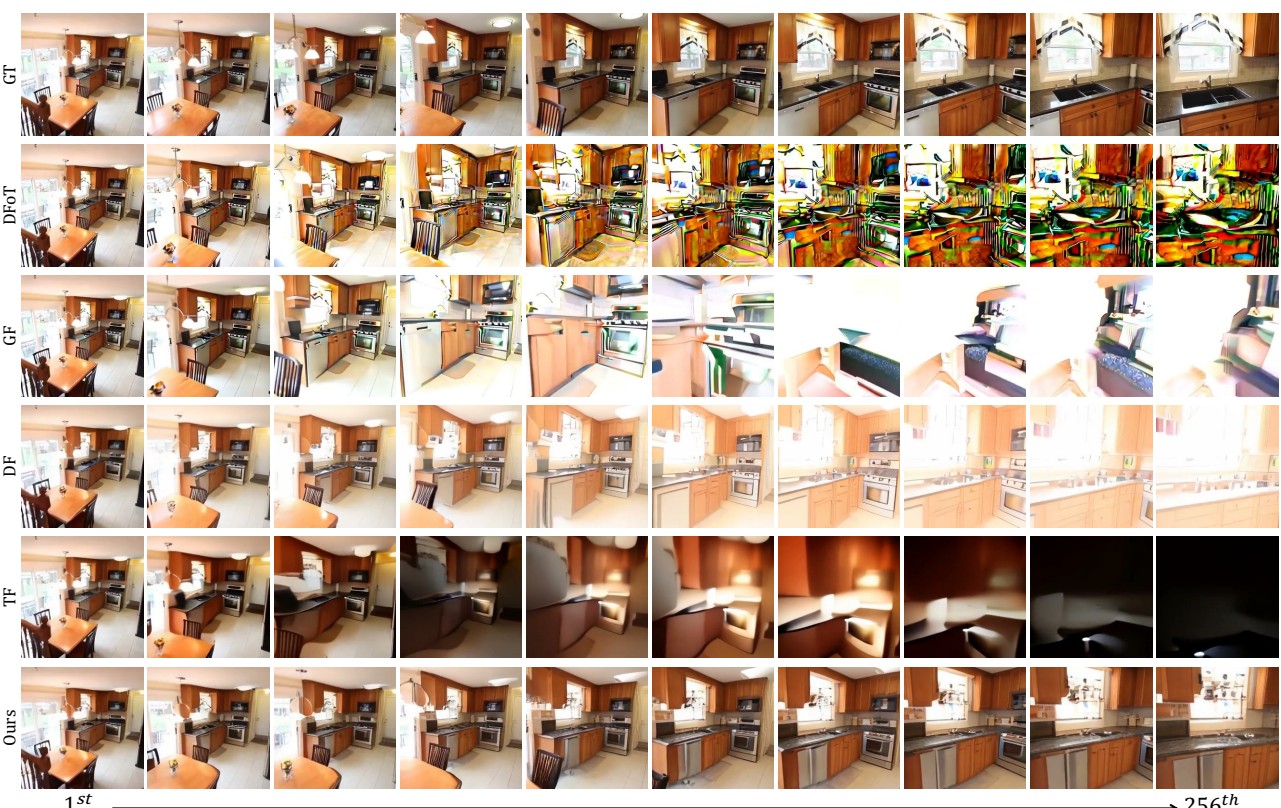

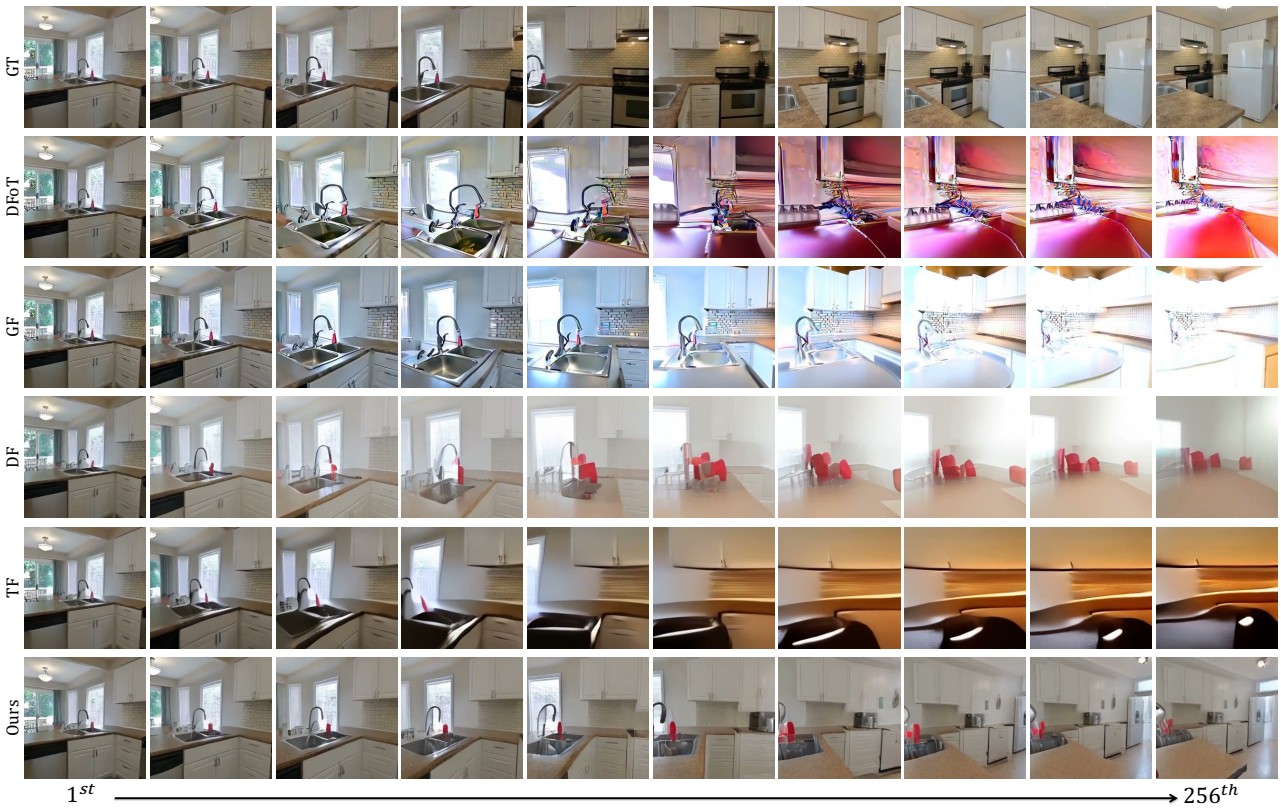

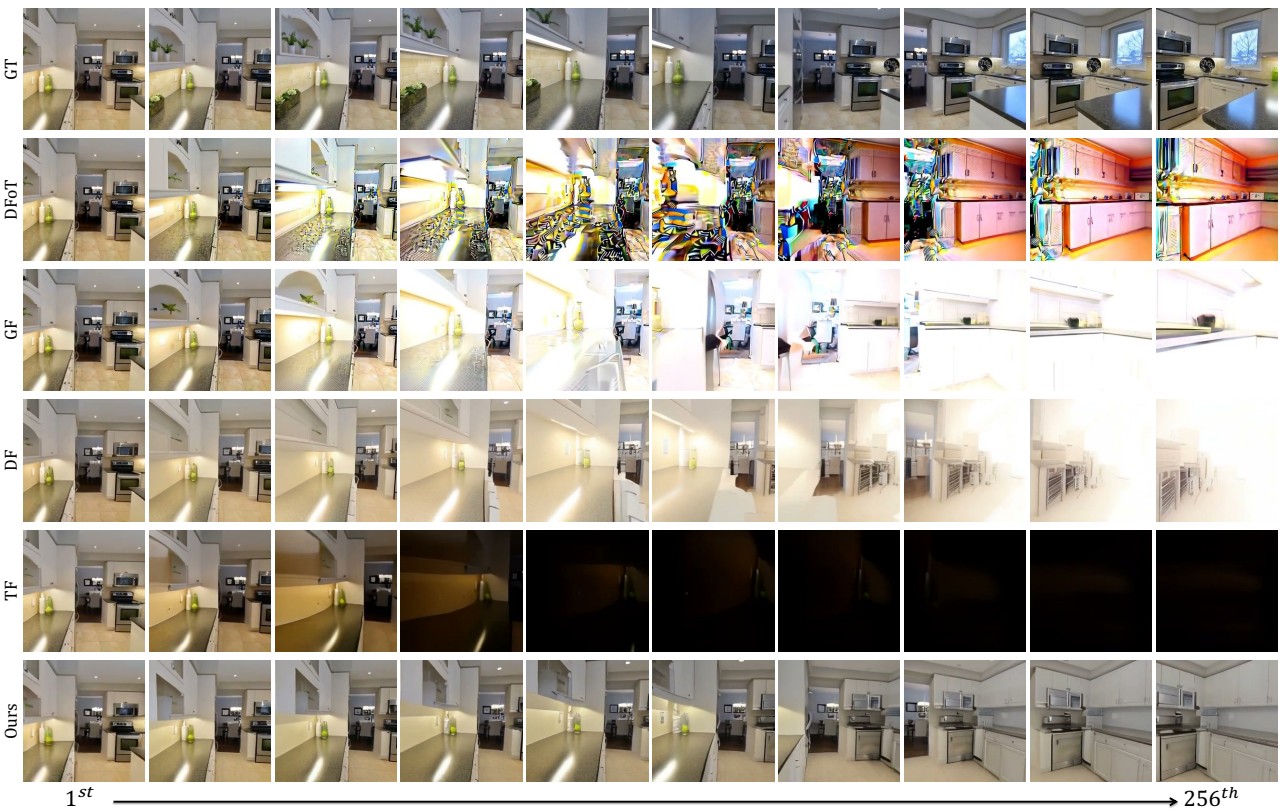

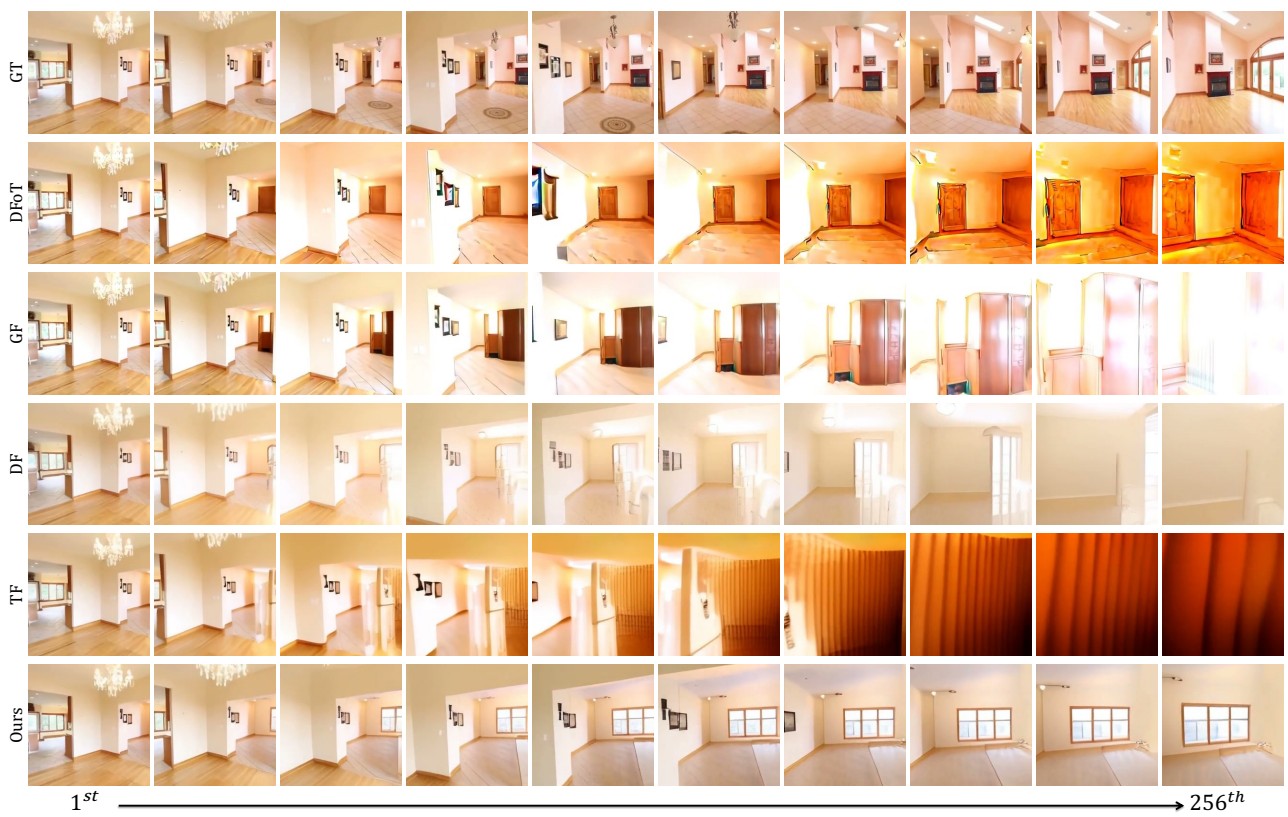

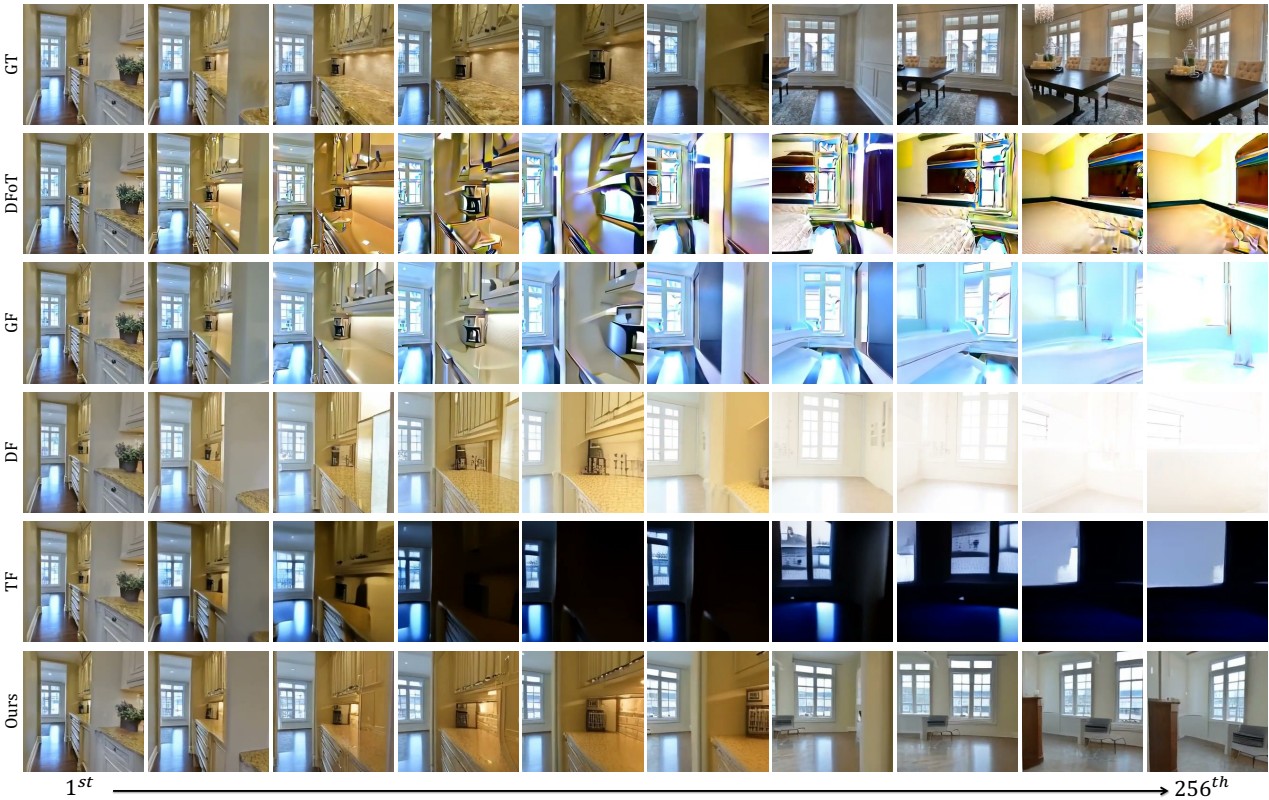

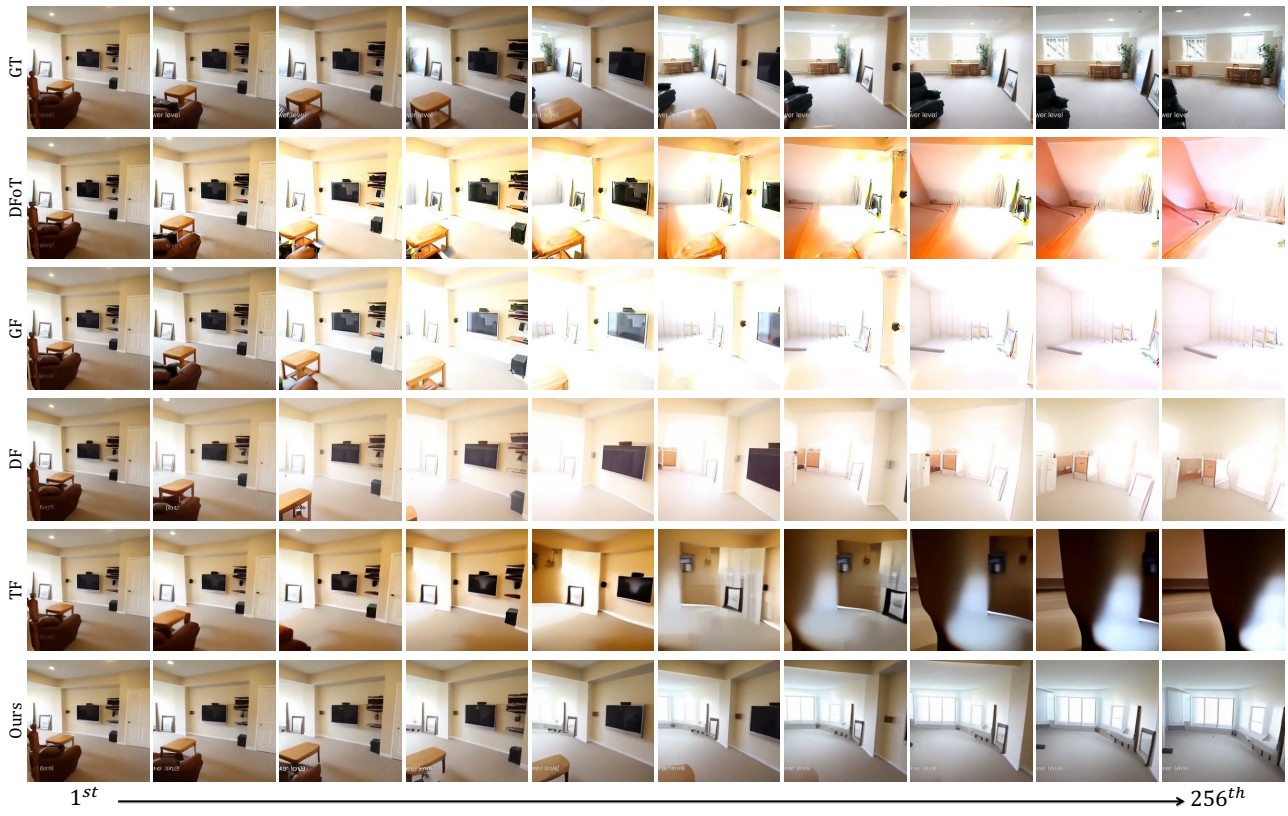

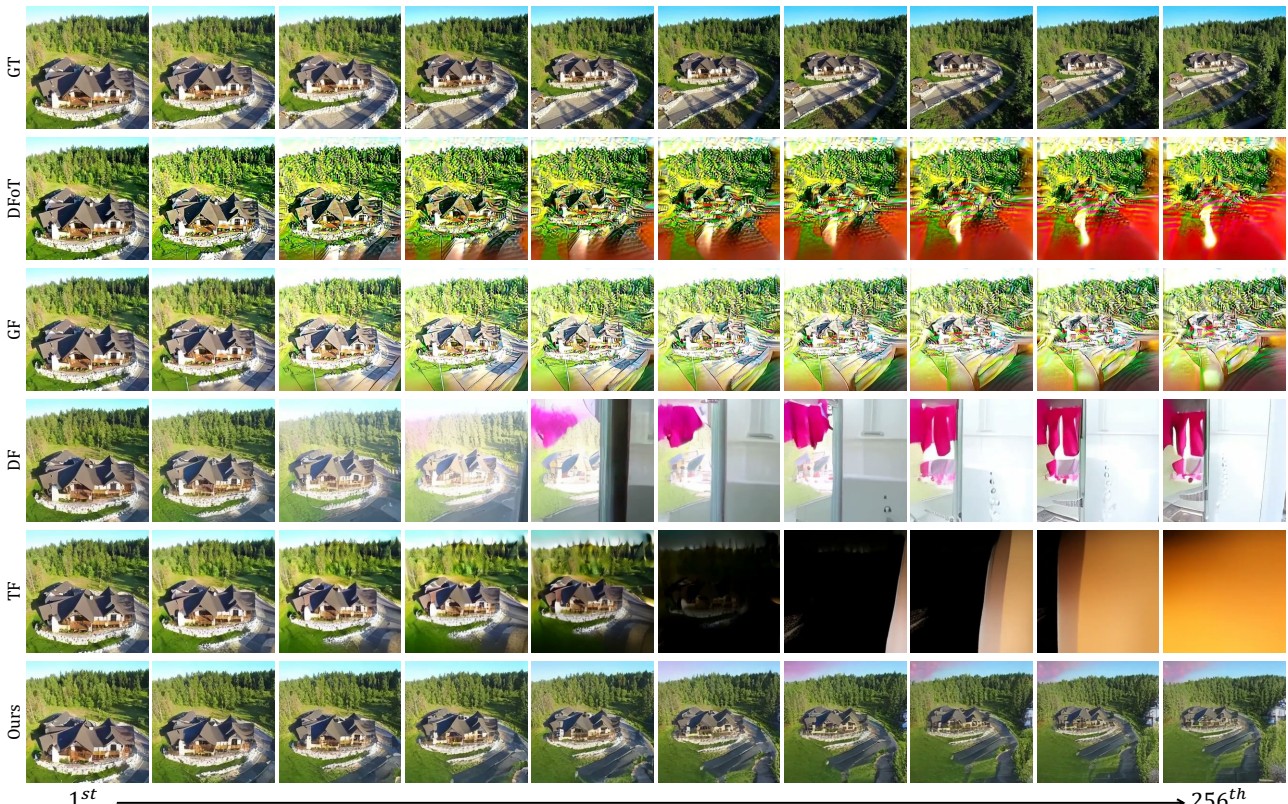

