# OpenReview forum: "LIVE: Long-horizon Interactive Video World Modeling"
_ICML.cc/2026/Conference — ICML 2026 regular_

### Official Review · Reviewer_qoS3 · 2026-03-08

**Soundness:** 3
**Presentation:** 2
**Significance:** 3
**Originality:** 3
**Overall Recommendation:** 4
**Confidence:** 4

**Summary:**

This paper proposes LIVE, an approach for long-horizon interactive video world modeling. The method introduces a novel cycle-consistency objective and a progressive training curriculum. Experiments on both real-world and gaming environments demonstrate that LIVE outperforms existing methods. In particular, LIVE is able to generate high-quality predictions over hundreds of steps.

**Compliance With Llm Reviewing Policy:**

Affirmed.

**Final Justification:**

The technical details of the proposed method become clearer, which resolves my major concerns.

**Key Questions For Authors:**

1. Technical details and intuition behind LIVE:
   - In Eq. (10), how are camera/action conditions reversed? Do you explicitly translate actions to their opposites (e.g., moving left vs. moving right), or do you simply reverse the action sequence? Does this require the action space to be invertible?
   - The notation in Eqs. (10–12) is difficult to follow, particularly due to the multiple superscripts (e.g.,  $\tilde{x}^{<k, \mathrm{rev}, \epsilon}$).
     - How do you extend supervision to the full window length $T$ by repeating $p$ frames? Does this imply that `T%p==0`?
     - How is the attention mask in Figure 4 constructed? For example, why does $x_{\mathrm{t}_7}^1$ attend to $x_{\mathrm{t}_3}^2$ but not to $x_{\mathrm{t}_5}^2$?
     - Is there any supervision for $\tilde{x}_{\mathrm{t}_1}^4$? Figure 4 shows a ground-truth frame $x^4$, but it appears visually quite different with $\tilde{x}_{\mathrm{t}_1}^4$, making the training signal unclear.
     - Does the model is trained for reverse order prediction ($x^4\rightarrow x^3\rightarrow x^2\rightarrow x^1$) during the all training phases?
   - What exactly is the “shortcut” mentioned in Lines 272-273?
   - In Line 294, the paper states that gradient optimization maintains  $\mathcal{D}\left(x^k, \tilde{x}^k\right)$. However, Eq. (9) suggests that generation is performed with gradients disabled. Could the authors clarify how gradients propagate in this case?
2. In this paper, DFoT performs poorly when predicting hundreds of steps. However, in the original DFoT paper, the method is able to generate high-quality results over much longer horizons. Could the authors explain the discrepancy?
3. It would be helpful to add x-axis labels in Figure 6, as it is currently unclear what the values (0, 20, 40, 60) represent.

Given the potential high impact of this work, I currently lean toward rejection because its curent presentation may be difficult for many readers to follow. However, if the authors can clarify these issues and improve the presentation, I would be willing to significantly increase my rating.

**Limitations:**

yes

**Strengths And Weaknesses:**

Strengths

1. Comprehensive evaluation and strong performance. The visual quality of the predictions over hundreds of steps is particularly impressive.
2. The proposed training approach appears generally reasonable and relatively easy to implement compared with more complex distillation-based methods. This makes the method practical and potentially highly valuable for this field.

Weaknesses:

1. The presentation is difficult to follow in several places, particularly regarding the key algorithmic component—the cycle-consistency objective—and the intuition behind it (see questions below).

---

> ### Author Rebuttal · Authors · 2026-03-31
>
> We appreciate your positive and constructive feedback. We are glad that our method is recognized for its comprehensive evaluation, strong performance, and potential highly value to the field. We respond to the questions below and provide qualitative Minecraft results at: https://anonymous.4open.science/w/QS3/
>
> >**KQ1-1: Reversing camera/action conditions and the invertibility requirement.**
>
> In our original setting, the action space is required to be invertible, so we kept only reversible controls (movement, camera rotations), with the reverse process implemented by reversing the video/action sequence and applying inverse controls (e.g., left $\rightarrow$ right).
>
> Motivated by your question, we extended LIVE beyond strictly reversible controls and tested it on irreversible Minecraft actions such as breaking blocks or consuming items (results in the [link](https://anonymous.4open.science/w/QS3/)). We introduce a binary condition (0/1 for forward/backward), embedded and added to the action embedding. During backward training, we reverse the video and action sequences, set the condition to 1, and mix with standard forward training at a 50% ratio (consistent with our original setup). This enables both forward prediction and backward retrodiction in the original action space.
>
> >**KQ1-2: Clarifications on notation, 1) repeated-prompt supervision, 2) attention-mask construction, 3) supervision targets, and 4) reverse-order training.**
>
> We apologize for the lack of notational clarity and will simplify it in the revision.
>
> 1. For example, if T=7 and p=3, the prompt frame ($x_1$, $x_2$, $x_3$) is extended to ($x_3$, $x_3$, $x_2$, $x_2$, $x_1$, $x_1$, $x_1$). Since the repeated prompt can simply be truncated at the end, T does not need to be divisible by p.
>
> 2. In Figure 4, the attention mask is constructed so that denoising frames ($x\_{t_5}\^2$ to $x_{t_8}\^1$) can attend only to conditioning frames ($\tilde{x}\_{t_1}\^4$ to $x\_{t_4}\^1$). This keeps a unified masking rule for different values of p. Hence, $x\_{t_7}\^1$ can attend to $x\_{t_3}\^2$ but not $x\_{t_5}\^2$.
>
> 3. There is no direct supervision on $\tilde{x}_{t_1}^4$ itself. The reason is exactly the semantic inconsistency highlighted in Fig. 3 of the paper. In LIVE the rollout frames are used only as reversed conditioning context, while supervision is applied to the repeated GT prompt targets. The red dashed boxes in Figure 4 indicate the frames where the loss is applied. We will add this label and clarify in the revision.
>
> 4. The model is exposed to reverse order throughout all training phases. 50% of the training videos are deliberately temporally reversed at data loading time, so for those samples the reversed video is treated as the original temporal order by the model.
>
> >**KQ1-3: Meaning of the "shortcut".**
>
> By “shortcut,” we mean the following degenerate case: because forward rollout quality typically degrades over time, after reversal the context quality improves monotonically. Then, when recovering $x_1$, the model may rely primarily on the nearest high-quality reversed frame (e.g., $x\^{2}$) and trivially satisfy the recoverability objective. We therefore inject random per-frame noise to prevent this behavior.
>
> >**KQ1-4: About gradient propagation.**
>
> In Eq. (9), Step 1 performs forward rollout with gradients disabled, while Step 2 updates the model through Cycle-consistency loss. Therefore, the model parameters are updated at each iteration and propagate indirectly to optimize forward rollout quality and help maintain $\mathcal{D}\left(x^k, \tilde{x}^k\right)$ across training iterations. We will clarify this more explicitly in the revision.
>
> >**KQ2: About the discrepancy with DFoT.**
>
> The discrepancy mainly comes from the generation protocol. In our paper, we evaluate strict long-horizon autoregressive rollout, where every future frame is generated sequentially and fed back as context. In contrast, the results in the original DFoT work are obtained under a different protocol: only sparse keyframes are generated autoregressively, and the remaining frames are filled in using bidirectional interpolation. We will clarify this distinction more explicitly in the revision.
>
>  >**KQ3: About the x-axis labels in Figure 6.**
>
> The x-axis values (0, 20, 40, 60) correspond to validation checkpoints, i.e., training steps (0, 20 $\times$ 500, 40 $\times$ 500, 60 $\times$ 500). We have updated Figure 6 accordingly, and the revised figure is provided in the [link](https://anonymous.4open.science/w/QS3/).

---

> > ### Author Rebuttal · Reviewer_qoS3 · 2026-04-02
> >
> > I thank the authors for the response. Now the technical details of the proposed method become clearer. I will increase my score.

---

> > > ### Author Response · Authors · 2026-04-07
> > >
> > > Thank you for your careful reading and helpful comments. We are grateful for your feedback and glad that our rebuttal helped clarify the paper; we will incorporate the above discussions and clarifications into the revised manuscript.

---

### Official Review · Reviewer_m9hq · 2026-03-12

**Soundness:** 2
**Presentation:** 3
**Significance:** 3
**Originality:** 3
**Overall Recommendation:** 4
**Confidence:** 3

**Summary:**

This article proposes a novel video world model framework called LIVE, aimed at solving the problem of error accumulation in long-term prediction generated by autoregression. By introducing the 'Cycle Consistency Objective', the model calculates diffusion loss through a closed-loop computation of forward generation and backward reconstruction without relying on a large pretrained teacher model, effectively and explicitly limiting the error propagation of long sequences. In addition, the author proposes a progressive training curriculum that unifies previous forced training methods and further enhances the training stability of the model.

**Compliance With Llm Reviewing Policy:**

Affirmed.

**Final Justification:**

The authors have almost addressed my previous concerns, so I would keep my positive view to this paper as 4.

**Key Questions For Authors:**

1.Robustness of Cycle-Consistency in Highly Unstructured Environments: The model demonstrates impressive performance on datasets like RealEstate 10K, Minecraft, and UE Engine. However, how resilient is the reverse generation process (cycle-consistency) in highly chaotic or unstructured environments (e.g., navigating through dense rubble or disaster zones)? Would erratic camera ego-motion or severe occlusions cause the reverse mapping to fail or diverge?
2.Integration with 3D Spatial Representations: The framework excels at generating long-horizon 2D video observations. Given that interactive agents ultimately require 3D spatial awareness, how easily could LIVE's autoregressive predictions be coupled with underlying 3D structures、to support downstream spatial planning?
3.Generalization of the Progressive Training Curriculum: The progressive training curriculum relies on manually adjusting the ratio of ground-truth frames to model-generated rollouts. Does this scheduling require heavy empirical tuning when transitioning to environments with drastically different temporal dynamics, or is the decay schedule generally transferable?

**Limitations:**

1. The exceptionally high demand for computing resources significantly impedes reproducibility for broader research communities. More critically, this computational bottleneck restricts the model's potential deployment on edge devices or autonomous mobile platforms where onboard compute power is strictly limited, making real-time, long-horizon inference impractical outside of server-grade hardware.
2. he dependency on clean, deterministic control signals drastically limits the model's practical application scenarios. In highly unstructured and chaotic environments—such as post-disaster search and rescue operations—agents often suffer from noisy odometry, erratic camera ego-motion, and a lack of predictable control inputs. Under such unpredictable physical dynamics, the model's autoregressive generation is highly susceptible to divergence or catastrophic failure.

**Strengths And Weaknesses:**

Strengths:
1.	A novel cyclic consistency training objective has been proposed, which effectively solves the problem of error accumulation in autoregressive video generation without relying on teacher model distillation.
2.	Unified Teacher Forcing and Diffusion Forcing into the same framework, and designed a progressive training course with solid theoretical contributions.
3.	The experimental design is sufficient, and SOTA results have been achieved on multiple long video generation benchmarks, especially with significant performance improvements on long sequences (≥ 256 frames).
4.	The complete ablation experiment validated the effectiveness of each module, and the qualitative results clearly demonstrated the improvement in generation quality.

Weaknesses:
1.	The model has 774M parameters and requires 32 H100 GPUs for training, which results in high computational costs and may limit its application in resource constrained scenarios.
2.	The main experiments focus on scenes with controllable camera trajectories (such as RealEstate 10K) and game environments, and the generalization ability to open world, highly dynamic real-world scenes needs to be verified.
3.	Circular consistency training requires reversing action/camera conditions, which may limit its applicability in scenarios where precise control signals cannot be obtained (such as real videos).

---

> ### Author Rebuttal · Authors · 2026-03-31
>
> Thank you very much for the great review helping us improve the work.  We are glad that our method is recognized as novel, with solid theoretical contributions and sufficient experimental validation. We respond to the questions below and provide qualitative SpatialVID results at: https://anonymous.4open.science/w/SVid/
>
> >**W1 & L1: Concerns about training cost, reproducibility, and deployment feasibility.**
>
> We verified full reproducibility using 8 GPUs (batch size 8×8) in ~30 hours, or 4 GPUs (batch size 4×8, 2-step gradient accumulation) in ~49 hours. The 32-H100 setup was used to reduce training time, not a strict requirement of LIVE. Moreover, LIVE only modifies training and adds no inference cost over the baseline (NFD, >20 FPS on a single RTX 4090).
>
> NFD: *Playing with Transformer at 30+ FPS via Next-Frame Diffusion*.
>
> >**W2 & W3 & KQ1 & L2: Applicability and robustness of LIVE under noisy / weak / non-ideal control signals in highly unstructured environments, including open-world and highly dynamic real-world scenes.**
>
> We further trained and evaluated LIVE on SpatialVID, a large-scale **real-world** video dataset with diverse scenes, strong dynamics, and challenging motion trajectories, including crowded streets with dense pedestrians/vehicles, visually complex outdoor scenes like jungles, as well as scenarios with extreme weather and severe occlusions. Here, the backward signal is injected via prompt modification (e.g., prepending "a reversed video of"), and camera information is estimated from video using ViPE rather than precise manual annotation. Despite much noisier conditions and weaker control signals, LIVE still consistently outperforms strong baselines across VBench metrics on a 5,000-video test set (Table C1). Qualitative examples are provided in the [link](https://anonymous.4open.science/w/SVid/).
>
> **Table C1: Quantitative comparison of video generation consistency and quality on the Spatial-VID dataset using VBench metrics.**
>
> ---
> | Method | Base Model | Teacher Model | Subject Consistency | Background Consistency | Temporal Flickering | Motion Smoothness | Aesthetic Quality | Imaging Quality | Overall Consistency |
> |:---|:---|:---|:---|:---|:---|:---|:---|:---|:---|
> | **Self-Forcing** | Wan&nbsp;2.1&nbsp;1.3B | Yes&nbsp;(14B) | 0.9198 | 0.9144 | 0.9489 | 0.9790 | 0.5495 | 69.2581 | 0.2498 |
> | **Matrix-Game&nbsp;2.0** | Wan&nbsp;2.1&nbsp;1.3B | Yes&nbsp;(14B) | 0.8831 | 0.9200 | 0.9303 | 0.9727 | 0.4976 | 67.5635 | 0.2152 |
> | **LIVE (Ours)** | Wan&nbsp;2.1&nbsp;1.3B | No | **0.9631** | **0.9665** | **0.9705** | **0.9870** | **0.6043** | **73.0653** | **0.2551** |
> ---
>
>
> More broadly, LIVE has now been validated across real and simulated environments, indoor and outdoor scenes, static and dynamic scenarios, settings with and without explicit control signals, and game videos.
>
> >**KQ2: Integration with 3D Spatial Representations.**
>
> LIVE and 3D spatial representations are complementary. LIVE focuses on generating consistent long-horizon 2D video predictions, which could be lifted into 3D representations (e.g., via depth estimation or neural radiance fields) to support spatial planning tasks such as navigation and manipulation. Further incorporating 3D spatial structure could also benefit long-term consistency and enable LIVE to serve as a more effective simulator for downstream planning modules.
>
> >**KQ3: Generalization of the Progressive Training Curriculum.**
>
> The progressive curriculum gradually shifts the context distribution from GT frames to model-generated rollouts, rather than introducing an environment-specific schedule. In practice, we find it broadly transferable: the same decay ratio r=4 is used across real and simulated environments, indoor/outdoor scenes, and game domains. We also tested r=2 and obtained similar results (see table below), suggesting the schedule is robust and does not require heavy empirical tuning.
>
> **Table C2: Quantitative results on RealEstate10K dataset.**
>
> ---
> | Method (LIVE) | PSNR (64f) | LPIPS (64f) | SSIM (64f) | PSNR (128f) | LPIPS (128f) | SSIM (128f) | PSNR (200f) | LPIPS (200f) | SSIM (200f) | PSNR (≥256f) | LPIPS (≥256f) | SSIM (≥256f) |
> | :--- | :---: | :---: | :---: | :---: | :---: | :---: | :---: | :---: | :---: | :---: | :---: | :---: |
> | Ratio=4 | 18.11 | 0.2215 | 0.5810 | 15.91 | 0.3298 | 0.5096 | 14.57 | 0.4163 | 0.4630 | 13.89 | 0.4682 | 0.4400 |
> | **Ratio=2** | **18.10** | **0.2219** | **0.5809** | **15.88** | **0.3291** | **0.5090** | **14.56** | **0.4152** | **0.4633** | **13.88** | **0.4689** | **0.4388** |
> ---

---

> > ### Author Rebuttal · Reviewer_m9hq · 2026-04-01
> >
> > The authors have almost addressed my previous concerns, so I would keep my positive view to this paper.

---

> > > ### Author Response · Authors · 2026-04-07
> > >
> > > Thank you for your positive review and valuable suggestions. We sincerely appreciate your feedback, which helped us improve the paper, and we will incorporate the above discussions and clarifications into the revised manuscript.

---

### Official Review · Reviewer_sQHD · 2026-03-12

**Soundness:** 3
**Presentation:** 3
**Significance:** 3
**Originality:** 3
**Overall Recommendation:** 5
**Confidence:** 3

**Summary:**

The paper proposes a training method for autoregressive video world models that uses a cycle-consistency objective to fight error accumulation over long rollouts. Instead of distilling from a teacher, the model rolls forward from GT prompt frames, reverses the rollout temporally and then tries to recover the original prompts via diffusion loss. Progressive curriculum reduces the number of GT prompt frames over training. Results on RealEstate10K, UE Engine, and Minecraft show improved metrics at long horizons vs TF, DF, and other baselines.

**Compliance With Llm Reviewing Policy:**

Affirmed.

**Final Justification:**

LIVE trains video world models to recover their own rollouts back to ground truth, which keeps errors from raising over long horizons. The idea is simple, doesn't slow down inference, and works well across three benchmarks. The rebuttal filled the gaps in the original submission. I support acceptance.

**Key Questions For Authors:**

1. How are actions reversed for minecraft specifically? can you ablate the cycle objective on ninecraft alone to show it actually helps when reversibility is approximate?
2. Why is FVD not reported anywhere? seems a natural metric for this task

**Limitations:**

yes

**Strengths And Weaknesses:**

Strengths:
1. the cycle-consistency idea for getting supervision on self-generated rollouts without a teacher is interesting and motivated
2. ablations in table are clean - each component matters and effects are directionally consistent
3. training cost is low - just 20k iters on top of a converged DF checkpoint for Re10K
4. the method is simple to implement - no adversarial loss or external models needed

Weakness:
1. the whole thing requires reversible actions/camera poses. for RealEstate10K this is fine but for Minecraft many actions are irreversible (e.g. breaking blocks or consuming items). authors don't discuss this and I suspect the reverse trajectory is physically implausible in many cases, which would make the cycle signal noisy or meaningless.
2. method is initialized from a converged DF checkpoint, so the gain could just be that the model is ready for any kind of on-policy finetuning after DF saturates. Figure 6 shows continued DF stagnates but thats "same objective vs new objective", not "this objective vs other plausible self-play objectives." hard to tell if the cycle loss specifically matters or if something simpler would work too.
3. FVD metric is completely absent

---

> ### Author Rebuttal · Authors · 2026-03-31
>
> We appreciate your positive and constructive feedback. We are glad that our method is recognized as interesting and well motivated, with clean ablations, and practical simplicity. We respond to the questions below and provide qualitative minecraft results at: https://anonymous.4open.science/w/sQHD/
>
> >**W1 & KQ1: About reversibility in Minecraft and how LIVE handles irreversible actions.**
>
> In our original Minecraft setting, we filtered out irreversible actions and kept only reversible controls (movement, camera rotations), where the reverse process is implemented by reversing the video/action sequence with corresponding inverse controls (e.g., left $\rightarrow$ right).
>
> Motivated by your question, we extended LIVE to handle irreversible actions such as breaking blocks or consuming items. We introduce a binary condition (0/1 for forward/backward), embedded and added to the action embedding. During backward training, we reverse the video and action sequences, set the condition to 1, and mix with standard forward training at a 50% ratio. This enables both forward prediction and backward retrodiction even when the reverse trajectory is physically implausible. The broader motivation is that a world model should not only predict the future but also retrodict the past, and LIVE shifts the conditioning distribution from GT context toward the model's own rollouts in both temporal directions.
>
> This extension works exceptionally well. Demos of backward retrodiction under irreversible actions are provided in the [link](https://anonymous.4open.science/w/sQHD/).
>
> >**KQ1: Minecraft-only ablation of the cycle objective under approximate reversibility.**
>
> We provide the Minecraft-only ablation results in the table below. The results show that the cycle objective remains effective on Minecraft.
>
> **Table B1: Effect of the cycle-consistency objective.**
>
> | Variant | PSNR (0-64f) | LPIPS (0-64f) | SSIM (0-64f) | PSNR (0-200f) | LPIPS (0-200f) | SSIM (0-200f) |
> |:---|:---|:---|:---|:---|:---|:---|
> | w/o Cycle | 14.58 | 0.4534 | 0.5129 | 12.98 | 0.5345 | 0.4423 |
> | **LIVE (Ours)** | **16.31** | **0.3271** | **0.6291** | **14.02** | **0.4299** | **0.5885** |
>
> >**W3 & KQ2: Missing FVD evaluation.**
>
> We have now added FVD results in the table below, which are consistent with and further support the main conclusions of the paper. In the original submission, Figure 1 already reports the FID of different methods across different generation horizons and shows the long-horizon stability advantage of LIVE. We will include the added FVD results in the revised paper.
>
> **Table B2: Fréchet Video Distance (FVD) evaluation on the RealEstate10K dataset.**
>
> | Method | 128 Frames | 200 Frames | 256 Frames |
> |:---|:---|:---|:---|
> | CameraCtrl | 253.23 | 461.59 | 781.59 |
> | DFoT | 335.72 | 488.64 | 660.02 |
> | Geometry Forcing | 192.92 | 315.28 | 472.38 |
> | Teacher Forcing | **163.12** | 267.34 | 440.81 |
> | Diffusion Forcing | 187.73 | 242.46 | 376.36 |
> | **LIVE (Ours)** | 207.54 | **215.83** | **269.62** |
>
> **Table B3: Fréchet Video Distance (FVD) evaluation on the interactive game environment (UE Engine).**
>
> | Method | 128 Frames | 256 Frames | 400 Frames |
> |:---|:---|:---|:---|
> | Teacher Forcing | **137.56** | 231.23 | 323.02 |
> | Diffusion Forcing | 242.01 | 395.90 | 534.74 |
> | **LIVE (Ours)** | 157.62 | **195.76** | **249.09** |
>
>
> **Table B4: Fréchet Video Distance (FVD) evaluation on the Minecraft dataset.**
>
> | Method | 64 Frames | 128 Frames | 200 Frames |
> |:---|:---|:---|:---|
> | Teacher Forcing | 301 | 310 | 323 |
> | Diffusion Forcing | 221 | 243 | 265 |
> | **LIVE (Ours)** | **203** | **231** | **256** |
>
> >**W2: Whether LIVE’s gains come specifically from the cycle objective, rather than generic post-DF on-policy finetuning.**
>
> We implemented a simpler self-play baseline that continues training from the converged DF checkpoint by using the model’s own rollout as condition and the corresponding ground-truth future as target under the standard diffusion loss.
>
> **Table B5: Quantitative comparison against standard on-policy fine-tuning.**
>
> | Method | PSNR (0-64f) | LPIPS (0-64f) | SSIM (0-64f) | PSNR (0-200f) | LPIPS (0-200f) | SSIM (0-200f) |
> |:---|:---|:---|:---|:---|:---|:---|
> | Diffusion Forcing (Init) | 16.59 | 0.2558 | 0.5723 | 12.21 | 0.4956 | 0.4598 |
> | On-Policy FT (1500 steps) | 16.52 | 0.2632 | 0.5645 | 12.39 | 0.4959 | 0.4547 |
> | **LIVE (Ours)** | **18.11** | **0.2215** | **0.5810** | **14.57** | **0.4163** | **0.4630** |
>
> This simpler on-policy fine-tuning strategy does not work well (Table B5). The reason is exactly the semantic inconsistency highlighted in Fig. 3 of the paper. In contrast, LIVE avoids this issue by reconstructing back to the original prompt frame, which is naturally consistent. This result shows that the gain is not from generic post-DF on-policy fine-tuning, but specifically from the cycle-consistency-style recovery objective.

---

> > ### Author Rebuttal · Reviewer_sQHD · 2026-04-02
> >
> > The rebuttal addresses all three concerns. I raise my score.

---

> > > ### Author Response · Authors · 2026-04-07
> > >
> > > Thank you for your careful review and constructive comments. We greatly appreciate your feedback and are pleased that our rebuttal helped address your concerns, and we will incorporate the relevant clarifications and additional experimental results into the revised paper.

---

### Official Review · Reviewer_YCQv · 2026-03-15

**Soundness:** 3
**Presentation:** 4
**Significance:** 3
**Originality:** 3
**Overall Recommendation:** 5
**Confidence:** 4

**Summary:**

This paper introduces LIVE, a method for training autoregressive video world models that aims to reduce long-horizon error accumulation. The authors argue that existing approaches, such as Teacher Forcing and Diffusion Forcing, suffer from a train–test mismatch, and that teacher-distillation-based methods, such as Self Forcing, help but add computational overhead and still do not explicitly bound long-horizon drift.

The key idea of the paper is to replace direct supervision of forward rollouts with a cycle-consistency objective. Starting from p prompt frames, the model first rolls forward to generate the remaining frames T-p, then reverses that rollout and the corresponding actions or camera conditions, adds random per-frame noise, and trains to reconstruct the original prompt frames from this reversed, imperfect context. This gives a valid training signal even when the forward rollout diverges semantically from the single ground-truth future, and it implicitly encourages the rollout to stay within a recoverable error range.

Experimentally, LIVE is evaluated on RealEstate10K, UE Engine Videos, and Minecraft, and the paper reports stronger long-horizon performance than prior baselines, especially at longer rollout lengths. On RealEstate10K, LIVE outperforms both TF and DF across all reported horizons, and the authors also show that LIVE maintains a stable FID as rollout length increases, whereas other baselines degrade more sharply. Ablation studies further suggest that all three pieces of the method matter: the cycle-consistency objective, the random context-noise strategy, and the progressive curriculum. Overall, the paper’s contribution is a training framework that improves long-horizon stability in interactive video world models without relying on teacher-based distillation.

**Compliance With Llm Reviewing Policy:**

Affirmed.

**Final Justification:**

The rebuttal has addressed my concerns. I moved my rating from weak accept to accept.

**Key Questions For Authors:**

- comparison to self-forcing

**Limitations:**

yes

**Strengths And Weaknesses:**

Soundness

- This is a well-executed paper on an important problem: reducing long-horizon error accumulation in interactive video models. The method is conceptually clean and practically appealing: LIVE replaces direct forward supervision with a cycle-consistency-style recovery objective while retaining the same base architecture and inference procedure as next frame diffusion (NFD), making the empirical gains easier to attribute to the training method itself. The experimental results are convincing overall, with consistent improvements on RealEstate10K, UE Engine, and Minecraft, and the ablations support the importance of the main components, including cycle consistency, context noise, and the progressive curriculum.

-  The main weakness is that the “bounded error accumulation” argument is more intuitive than formally established, and the paper does not compare against the strongest teacher-distillation/self-forcing style baselines, which slightly limits the strength of the empirical claims.


Presentation

- The paper is well structured. The motivation is easy to follow, the figures are helpful, and the unified view connecting TF, DF, and LIVE through the GT ratio p is especially nice. That said, a few implementation details---particularly the repeated-prompt supervision---could be explained more concretely, perhaps with a simple toy example.

Significance

- Long-horizon robustness is a central challenge in autoregressive video modeling, and this work offers a practical training improvement that does not require changing the inference pipeline. That makes the method both scientifically relevant and potentially useful for practitioners working on interactive simulation and video world models.

Originality

- The individual ingredients are not entirely new, but the overall formulation is creative and well-motivated. The idea of supervising recoverability rather than exact forward matching is a meaningful conceptual contribution, and the combination of forward rollout, reverse recovery, and diffusion-based training is novel enough to stand out.

---

> ### Author Rebuttal · Authors · 2026-03-31
>
> We appreciate your positive and constructive feedback. We are encouraged that the paper is viewed as a well-executed study on an important problem, with a creative and well-motivated formulation. We respond to the questions below and provide qualitative SpatialVID results at: https://anonymous.4open.science/w/SVid/
>
> >**W1 & KQ1: Comparison with Self-Forcing as empirical support for bounded-error behavior**
>
> We agree that this bounded-error behavior is not explicitly derived theoretically but empirically supported by our long-horizon results where LIVE exhibits nearly no FID degradation (Figure 1 in the main text).
>
> Following your suggestion, we added stronger comparisons against Self Forcing and the Self-Forcing-based world model Matrix-Game 2.0 on SpatialVID using VBench metrics. In these comparisons, LIVE uses the same base model as Self Forcing / Matrix-Game 2.0, namely Wan2.1-1.3B, while not requiring distillation from a 14B teacher model. The new results below further strengthen the empirical support for our claims.
>
> **Table A1: Quantitative comparison of video generation consistency and quality on the Spatial-VID dataset using VBench metrics. We compare LIVE against Self-Forcing and Matrix-Game 2.0, on a test split of 5,000 videos.**
>
> ---
> | Method | Base Model | Teacher Model | Subject Consistency | Background Consistency | Temporal Flickering | Motion Smoothness | Aesthetic Quality | Imaging Quality | Overall Consistency |
> |:---|:---|:---|:---|:---|:---|:---|:---|:---|:---|
> | **Self-Forcing** | Wan&nbsp;2.1&nbsp;1.3B | Yes&nbsp;(14B) | 0.9198 | 0.9144 | 0.9489 | 0.9790 | 0.5495 | 69.2581 | 0.2498 |
> | **Matrix-Game&nbsp;2.0** | Wan&nbsp;2.1&nbsp;1.3B | Yes&nbsp;(14B) | 0.8831 | 0.9200 | 0.9303 | 0.9727 | 0.4976 | 67.5635 | 0.2152 |
> | **LIVE (Ours)** | Wan&nbsp;2.1&nbsp;1.3B | No | **0.9631** | **0.9665** | **0.9705** | **0.9870** | **0.6043** | **73.0653** | **0.2551** |
> ---
>
> Matrix-game 2.0: *An open-source real-time and streaming interactive world model*.
>
> Self Forcing: *Bridging the Train-Test Gap in Autoregressive Video Diffusion*.
>
> SpatialVID: *A Large-Scale Video Dataset with Spatial Annotations*.
>
> >**W2: “That said, a few implementation details, particularly the repeated-prompt supervision, could be explained more concretely, perhaps with a simple toy example.”**
>
> A concrete toy example is a training window of length 6 with prompt length $p=2$. The clean prompt is $(x^1, x^2)$. In the reverse-generation setup, the full window contains both rollout context and supervision targets,  and the repeated-prompt supervision applies specifically to the target side. In our repeated-prompt supervision, we repeat the clean prompt across the full window, so the supervision targets become $(x^1, x^2, x^1, x^2, x^1, x^2)$. Each repeated frame then independently samples its own diffusion timestep and noise, e.g., $(x^1_{t_1}, x^2_{t_2}, x^1_{t_3}, x^2_{t_4}, x^1_{t_5}, x^2_{t_6})$. As a result, within one training pass, the same clean prompt contributes multiple timestep-conditioned supervision instances, which makes the diffusion training objective much more efficient to estimate. (By contrast, in a standard setup one would construct only one noisy instance of each target frame, e.g., ($x^1_{t_1}$, $x^2_{t_2}$.) We will add this example to the paper to make the repeated-prompt supervision mechanism explicit.

---

> > ### Author Rebuttal · Reviewer_YCQv · 2026-04-05
> >
> > Thanks for the answers. I am satisfied.

---

> > > ### Author Response · Authors · 2026-04-07
> > >
> > > Thank you for your thoughtful review and helpful feedback. We sincerely appreciate your time, and we will incorporate the additional results and clarifications into the revised manuscript.

---

### Decision · Program_Chairs · 2026-04-30

**Decision:**

Accept (regular)

**Comment:**

The paper proposes a training method for autoregressive video world models that uses a cycle-consistency objective to fight error accumulation over long rollouts. Instead of distilling from a teacher, the model rolls forward from GT prompt frames, reverses the rollout temporally and then tries to recover the original prompts via diffusion loss. Progressive curriculum reduces the number of GT prompt frames over training. Results on RealEstate10K, UE Engine, and Minecraft show improved metrics at long horizons vs TF, DF, and other baselines.